

# On the contribution of nocturnal heterogeneous reactive nitrogen chemistry to particulate matter formation during wintertime pollution events in Northern Utah

Erin E. McDuffie[1,2,3ξ], Caroline Womack[1,2], Dorothy L. Fibiger[1,2†], William P. Dube[1,2], Alessandro
Franchin[1,2], Ann Middlebrook[1], Lexie Goldberger[4‡], Ben H. Lee[4], Joel A. Thornton[4], Alexander Moravek[5],
Jennifer Murphy[5], Munkhbayar Baasandorj[6§], Steven S. Brown[1,3]

[1]Chemical Sciences Division, National Oceanic and Atmospheric Administration, Boulder, CO, USA
[2]Cooperative Institute for Research in Environmental Sciences, University of Colorado, Boulder, CO, USA
[3]Department of Chemistry, University of Colorado, Boulder, CO, USA
[4]Department of Atmospheric Science, University of Washington, Seattle, WA, USA
[5]Department of Chemistry, University of Toronto, Toronto, Canada
[6]Department of Atmospheric Sciences, University of Utah, Salt Lake City, UT, USA

[ξ]Now at: Department of Physics and Atmospheric Science, Dalhousie University, Halifax, NS, Canada
[†]Now at: California Air Resources Board, Sacramento, CA, USA
[‡]Now at: ARM Aerial Facility, Pacific Northwest National Laboratory, Richland, WA, USA
[§]Now at: Chevron Corporation, Houston, TX, USA

*Correspondence to*: Steven S. Brown (steven.s.brown@noaa.gov)

**Abstract.** Mountain basins in Northern Utah, including Salt Lake Valley (SLV), suffer from wintertime air pollution events
associated with stagnant atmospheric conditions. During these events, fine particulate matter concentrations ($PM_{2.5}$) can exceed
national ambient air quality standards. Previous studies in SLV have found $PM_{2.5}$ is primarily composed of ammonium nitrate
($NH_4NO_3$), formed from the condensation of gas-phase ammonia ($NH_3$) and nitric acid ($HNO_3$). Additional studies in several
western basins, including SLV, have suggested that production of $HNO_3$ from nocturnal heterogeneous $N_2O_5$ uptake is the
dominant source of $NH_4NO_3$ during winter. The rate of this process, however, remains poorly quantified, in part due to limited
vertical measurements above the surface, where this chemistry is most active. The 2017 Utah Winter Fine Particulate Study
(UWFPS) provided the first aircraft measurements of detailed chemical composition during SLV wintertime pollution events.
Coupled with ground-based observations, analysis of day and nighttime research flights confirm that $PM_{2.5}$ during wintertime
pollution events is principally composed of $NH_4NO_3$, limited by $HNO_3$. Here, observations and box-model analyses assess the
contribution of $N_2O_5$ uptake to nitrate aerosol during pollution events using the $NO_3^-$ production rate, $N_2O_5$ heterogeneous uptake
coefficient ($\gamma(N_2O_5)$), and production yield of $ClNO_2$ ($\phi(ClNO_2)$), which had medians of 1.6 µg m⁻³ hr⁻¹, 0.076, and 0.220,
respectively. While fit values of $\gamma(N_2O_5)$ may be biased high by a potential under-measurement in aerosol surface area, other fit
quantities are unaffected. Lastly, additional model simulations suggest nocturnal $N_2O_5$ uptake produces 3.9 µg m⁻³ of nitrate per
day, when considering the possible effects of dilution. This nocturnal production is sufficient to account for 86% of the daily
observed surface-level build-up of aerosol nitrate, though accurate quantification is dependent on modeled dilution and mixing
processes.



## 1 Introduction

Over 80% of Utah's population lives in counties that experience periods of elevated fine particulate matter ($PM_{2.5}$ < 2.5 μm in diameter) during the winter season (U.S. Census Bureau, 2018; Whiteman et al., 2014). In these counties, the highest levels have been limited to three northern valleys along the Wasatch Mountains, shown in Figure 1 (north to south: Cache Valley (Logan Non-

attainment area (NAA)), Salt Lake Valley (Salt Lake NAA), and Utah Valley (Provo NAA)). These valleys were designated by the U.S. EPA as "Moderate" non-attainment areas (NAA) in December 2009, with the Salt Lake and Provo areas reclassified from moderate to "Serious" in May 2017 (Utah Department of Environmental Quality). Elevated $PM_{2.5}$ concentrations in these regions impact public health and are associated with increases in emergency room visits for asthma (Beard et al., 2012). Short-term exposure has also been shown to increase the chance of triggering acute ischemic heart disease events by 4.5-6% per 10 μg m$^{-3}$ of

$PM_{2.5}$ in sensitive populations living in the Wasatch region (Pope et al., 2006; Pope et al., 2015).

Elevated wintertime $PM_{2.5}$ concentrations in these valleys typically correspond to multi-day events of high atmospheric stability (e.g. Whiteman et al., 2014; Silcox et al., 2012; Gillies et al., 2010; Wang et al., 2012; Green et al., 2015; Silva et al., 2007; Baasandorj et al., 2017), associated with large, synoptic-scale high-pressure systems that transit from west to east, simultaneously impacting multiple basins across the intermountain western U.S. (e.g. Reeves and Stensrud, 2009). Warm

temperatures aloft cause boundary layer stratification that reduces mixing and traps cold air and emissions near the surface, illustrated in Figure 2 and discussed further below. These events, termed persistent cold air pools (PCAPs), typically mix-out after 1-5 days but have been observed to persist for as long as 18 days (Whiteman et al., 2014). Similar meteorological patterns have been linked to wintertime $PM_{2.5}$ accumulation in basins across the western U.S. (e.g. Chen et al., 2012; Green et al., 2015). During past PCAP and pollution events in Utah, data from ground-based measurements in Salt Lake Valley (SLV) have reported day to

day build-up rates of total $PM_{2.5}$ mass in the range of ~6-10 μg m$^{-3}$ day$^{-1}$ (Baasandorj et al., 2017; Silcox et al., 2012; Whiteman et al., 2014) before plateauing after ~ 6 days into an event (Baasandorj et al., 2017). Average 24-hour concentrations reported during PCAP events between 2001 and 2016 have been as large as 40-80 μg m$^{-3}$ in Salt Lake (Baasandorj et al., 2017; Silcox et al., 2012) and Utah Valleys (Malek et al., 2006), and up to 132.5 μg m$^{-3}$ in Logan, Utah (Cache Valley) (Malek et al., 2006).

Previous ground-based studies have identified ammonium nitrate ($NH_4NO_3$) as the main component of $PM_{2.5}$ (70 - 80%

by mass) during PCAP events in all three Northern Utah Valleys (Silva et al., 2007; Hansen et al., 2010; Kuprov et al., 2014; Kelly et al., 2013; Long et al., 2003; Long et al., 2005a; Long et al., 2005b; Baasandorj et al., 2017). Ammonium nitrate formation is thermodynamically favorable under cold wintertime conditions from the equilibrium between gas-phase ammonia ($NH_3$) and nitric acid ($HNO_3$), shown in Reaction (R1) in Figure 2 (e.g. Kuprov et al., 2014; Nowak et al., 2012; Mozurkewich, 1993). $PM_{2.5}$ mitigation strategies that are based on control of these gas-phase species are expected to be more effective if the limiting reagent

and its sources can be identified. Both model- and observationally-informed ground-based analyses have suggested that $NH_4NO_3$ formation in Cache and Salt Lake Valleys is limited by the production of $HNO_3$ (Kuprov et al., 2014; Mangelson et al., 1997; Martin, 2006; Utah Division of Air Quality, 2014b, a, c; Franchin et al., 2018), though uncertainties remain in how this limitation may be impacted by temporal and spatial variations.

While $NH_3$ is directly emitted from agricultural sources, industrial processes, waste disposal, and automobile emissions

(Behera et al., 2013; Livingston et al., 2009), $HNO_3$ forms chemically in the atmosphere from the oxidation of $NO_x$ (=NO + $NO_2$), which in turn arises mainly from combustion emissions. There are two mechanisms by which this formation occurs, illustrated by reactions (R2 - R6) in Figure 2. The first is through daytime $NO_2$ oxidation by the hydroxyl radical (OH) (Figure 2, R2) and the second is through the nocturnal heterogeneous uptake of dinitrogen pentoxide ($N_2O_5$) (R6), which itself is a product of nocturnal



NO$_x$ oxidation (R3 – R5). The former is relatively more important during the summer (Brown et al., 2004) whereas the latter, the focus of this study, may be relatively more important in winter (e.g. Wagner et al., 2013) due to reduced OH concentrations, colder temperatures that favor N$_2$O$_5$ in its equilibrium with NO$_3$ (R5), and longer nights that allow more time for nocturnal reactions to

occur. The nocturnal heterogeneous production of HNO$_3$ is also expected to be largest in the residual layer (RL), due to the near surface accumulation of NO, which titrates O$_3$ (R3) and reacts with NO$_3$ (R7), the precursor to N$_2$O$_5$ (e.g. Brown and Stutz, 2012).

The role of this nocturnal reactive nitrogen chemistry in the formation of PM$_{2.5}$ has been considered in previous wintertime studies, though nocturnal, vertically-resolved measurements have been limited. Previous studies using ground and tower-based observations, as well as mid-morning aircraft vertical profiles have identified heterogeneous chemistry and subsequent morning transport from aloft as a major source of surface-level NH$_4$NO$_3$ in California's San Joaquin Valley (e.g. Brown et al.,

2006; Prabhakar et al., 2017; Pusede et al., 2016; Watson and Chow, 2002). Nocturnal heterogeneous chemistry has also been considered as a source for PM$_{2.5}$ in northern Utah (Baasandorj et al., 2017; Kuprov et al., 2014), though vertically resolved measurements have been limited to ground-based observations at different elevations along the Wasatch Mountains (Baasandorj et al., 2017). In an analysis of ground-based HNO$_3$ and PM$_{2.5}$ observations in SLV, Kuprov et al. (2014) suggested that daytime HNO$_3$ formation was dominant over the contribution from nocturnal heterogeneous chemistry. Baasandorj et al. (2017), however,

noted that ground-based measurements in this region may not capture the extent of heterogeneous chemistry aloft in the RL, which is expected to be distinct from the surface composition (e.g. Brown et al., 2007; Brown and Stutz, 2012; Stutz et al., 2004). Therefore, vertical gradients in NO$_x$ and oxidants could promote efficient HNO$_3$ and NH$_4$NO$_3$ formation aloft, which could contribute to enhanced surface-level PM$_{2.5}$ concentrations the following day. Regardless of altitude, the absolute contribution will depend on 1) the rate of NO$_3$ and N$_2$O$_5$ production, 2) the efficiency of N$_2$O$_5$ uptake onto aerosol ($\gamma$(N$_2$O$_5$)) and 3) the heterogeneous

production yield of HNO$_3$ relative to ClNO$_2$ ($\phi$(ClNO$_2$)) (Osthoff et al., 2008; Behnke et al., 1997). Net accumulation surface-level NH$_4$NO$_3$, however, also depends on mixing and dilution associated with growth of the convective boundary layer and mixing of the RL down to the surface during the following day. Quantification of these processes is a key component in designing effective mitigation strategies for Utah's wintertime air pollution and requires vertically resolved observations of chemical composition at night.

In this study, we present results from the Utah Winter Fine Particulate Study (UWFPS), which consisted of aircraft and ground-based observations throughout Cache, Salt Lake, and Utah Valleys during January and February 2017. This analysis focuses on data from 16 aircraft flights (5 at night) during two pollution events between 16 January and 1 February 2017. These flights were carried out in SLV, the most populated of the three Utah Non-Attainment Areas. An overview of PM$_{2.5}$ during winter 2016-2017 is presented in the first section. Ambient mixing ratios of total (gas and particle-phase) oxidized and reduced nitrogen

are shown in the second section to assess the limiting reagent to NH$_4$NO$_3$ aerosol formation, as well as its spatial and temporal trends. The final section presents upper-limit NH$_4$NO$_3$ production rate estimates from aircraft observations and results from a chemical box model that is fit to observations to calculate $\gamma$(N$_2$O$_5$), $\phi$(ClNO$_2$), and an estimated contribution of nocturnal heterogeneous chemistry to NH$_4$NO$_3$ formation in SLV. The contribution of nocturnal production relative to photochemically-driven NO$_2$ oxidation will have consequences for the development of effective mitigation strategies as day and nighttime

production processes may have different sensitivities to NO$_x$ emissions and VOC radical sources (Pusede et al., 2016; Womack et al., 2019), such that net sensitivities will be determined by the dominant formation mechanism.



## 2 Methods

### 2.1 UWFPS Campaign Overview and Instrumentation

The UWFPS campaign included both aircraft and ground-based measurements throughout Salt Lake, Cache, and Utah Valleys during January and February 2017 (Figure 1). A total of 23 research flights were conducted during both day and night with the

NOAA Twin Otter (TO) aircraft, equipped with aerosol and gas-phase instrumentation (summarized in Table 1) to probe the regional sources and formation mechanisms of $PM_{2.5}$. While flights were conducted over three valleys, the focus of this analysis will be on the more densely populated SLV, with relevant flight tracks highlighted in the right panel of Figure 1.

Briefly, the TO payload included gas-phase measurements of $NO_x$, $NO_2$, $NO_y$, and $O_3$ (1 Hz sample frequency) from a NOAA Cavity Ring Down Spectrometer (NOxCaRD) (Wild et al., 2014), $NH_3$ (1 Hz sample frequency) measurements from an

Aerodyne mid infrared absorption instrument (QC-TIDLAS) from the University of Toronto (Ellis et al., 2010), and $N_2O_5$, $HNO_3$, and $ClNO_2$ (1 Hz sample frequency) measured with an iodide Time-of-Flight Chemical Ionization Mass Spectrometer (I⁻TOF-CIMS) from the University of Washington (Lee et al., 2014; Lee et al., 2018). Accuracies for $NO_x$, $NO_2$ and $O_3$ were 5% and 12% for $NO_y$. with stated detection limits of 60 pptv (2σ) (Wagner et al., 2011; Wild et al., 2014) in the boundary layer. Gas-phase $NH_3$ was measured with a detection limit of 450 pptv (1s 3σ), as described in further detail by Moravek et al. (2019). Accuracy and

detection limits for $N_2O_5$, $ClNO_2$, and $HNO_3$ were similar to those reported from the same instrument deployed during the Wintertime INvestigation of Transport, Emissions, and Reactivity (WINTER) campaign (≤ 0.6 pptv (1s 1σ), 30%) (Lee et al., 2018). Non-refractory sub-micron aerosol composition (sampled every ~ 10 s) was measured with the NOAA Aerosol Mass Spectrometer (AMS) (Bahreini et al., 2009; Middlebrook et al., 2012) and aerosol size (sample every ~ 3 s) with a commercial Ultra-High Sensitivity Aerosol Spectrometer (UHSAS) (Brock et al., 2011). Average detection limits for AMS aerosol composition

were 0.04, 0.09, 0.33, 0.03, and 0.07 μg $sm^{-3}$ ($sm^{-3}$ refers to $m^3$ under standard conditions (1 atm and 273.15 K)) for particulate nitrate, ammonium, organics, sulfate, and chloride, respectively. Uncertainties were ~20% for all species (Franchin et al., 2018). Ambient temperature and pressure (1 Hz sample frequency) were measured with a commercial (Avantech) meteorological probe. The accuracy of the commercial UHSAS instrument was also expected to be similar to that used during WINTER (dry surface area density: ~34%).

Additional ground-based measurements used in this analysis include hourly $PM_{2.5}$, $NO_2$, $O_3$, and temperature from the Utah Department of Air Quality (UDAQ) instrumentation at the Hawthorne (HW) monitoring site (Figure 1). Total $PM_{2.5}$ mass was measured with a Thermo Scientific 1405-DF Dichotomous Ambient Air Monitor, $NO_2$ with a Teledyne API T200U Chemiluminescence detector, and $O_3$ with a Teledyne API T400 UV absorption spectrometer, all in accordance with EPA guidelines (Environmental Protection Agency, 2018). Select volatile organic compounds (VOCs) were also measured at the

University of Utah (UU) ground site by a Proton-Transfer Reaction Mass Spectrometer. Further information about the UWFPS campaign and aircraft and ground-based instrumentation can be found in additional publications (Franchin et al., 2018; UWFPS Science Team, 2018; Womack et al., 2019; Moravek et al., 2019).





### 2.2 Box Model

#### 2.2.1 Description

A zero-dimension chemical box model has been developed to simulate the nocturnal chemical evolution of an air parcel from sunset until the time of aircraft measurement. Extensive model details have been previously discussed in McDuffie et al. (2018b).

Briefly, the model forward-integrates the chemical mechanism (13 reactions, Table S1) starting 1.3 hours prior to sunset (see below), iteratively adjusting the initial concentrations of $O_3$ and $NO_2$, until the model-predicted concentrations of both are within 0.5% of the aircraft observations. Holding these initial concentrations constant, the model next adjusts the total heterogeneous loss rate constant of $N_2O_5$ ($k_{N_2O_5}$) and production rate constant of $ClNO_2$ ($k_{ClNO_2}$) until the model output simultaneously reproduces ambient nighttime observations of $N_2O_5$ and $ClNO_2$ to within 1%. The $N_2O_5$ uptake coefficients ($\gamma(N_2O_5)$) and $ClNO_2$ production

yields ($\phi(ClNO_2)$) are then calculated following Eqs. (1) and (2), where $c$ is the mean molecular speed and $SA$ is the ambient wet $PM_1$ surface area density (described below). The model repeats this process every 10 seconds for all flights conducted at night, as determined by time and aircraft GPS altitude.

$$\gamma(N_2O_5) = \frac{4 * k_{N_2O_5}}{c * SA} \tag{1}$$

$$\varphi(ClNO_2) = \frac{k_{ClNO_2}}{k_{N_2O_5}} \tag{2}$$

15        Holding the derived $k_{N_2O_5}$ and $k_{ClNO_2}$ values constant, the model can further simulate the total nitrate produced overnight by forward-integrating the model until the time of sunrise, as shown for a representative SLV point in Figure 3. Here, total nitrate (gas + particulate-phase) is represented as $HNO_3$ only, as this model does not include aerosol thermodynamics that partition nitrate between the gas and particle phases. Modeled gas-phase $HNO_3$ is assumed to partition to the particle phase with 100% efficiency, following observations presented in Franchin et al. (2018) that show > 90% of total nitrate is in the particle phase during wintertime

pollution events in SLV. As modeled nitrate is initialized with a concentration of 0 μg m⁻³, concentrations predicted at sunrise, therefore represent the total amount of nitrate produced from nocturnal chemistry over the course of a single night (i.e. nocturnal nitrate production rate). These base case values assume no overnight loss from dilution and constant values of $\gamma(N_2O_5)$ and $\phi(ClNO_2)$, as discussed further in Section 3.3.3.

#### 2.2.2 Model Simplifications and Uncertainties

For the UWFPS campaign, the box model was run in a similar manner to that described previously in McDuffie et al. (2018b), for nocturnal aircraft observations collected in the RL over the eastern U.S. coast during the 2015 WINTER campaign. Due to more limited instrumentation during UWFPS than WINTER, a larger number of box model assumptions and simplifications were required, which are summarized below.

First, the wet SA density for the base case simulations was calculated by applying a hygroscopic growth curve as a

function of RH (Figure S2) to the dry $PM_1$ SA measured by the UHSAS (details in Section S1.3). The growth curve was derived with the Extended-AIM Aerosol Thermodynamic Model (Wexler and Clegg, 2002), assuming pure $NH_4NO_3$ particles.



Alternatively, estimating the growth factor from AMS organic measurements and estimates of aerosol organic density and hygroscopicity constant ($\kappa_{Org}$) (described in S1.3, (Jimenez et al., 2009; Mei et al., 2013; Cerully et al., 2015; e.g. Kuwata et al., 2012; Brock et al., 2016; Shingler et al., 2016)), resulted in only a ~3% change in the total wet SA for night flights during UWFPS (Figure S2a). For the 1031 measurement periods with simultaneous values of $\gamma(N_2O_5)$ and $\phi(ClNO_2)$, the median dry aerosol SA was 151 $\mu m^2$ $cm^{-3}$, which increased to 353 $\mu m^2$ $cm^{-3}$ when accounting for hygroscopic growth (Figure S2b). Additional uncertainties associated with aerosol SA are discussed below in Section 3.3.2 and only impact the $\gamma(N_2O_5)$ values reported here (not the nitrate production rates) as $k_{N_2O_5}$ and $k_{ClNO_2}$ are independent of the aerosol SA.

Second, loss of the nitrate radical ($NO_3$) from its reaction with volatile organic compounds (VOCs) was assumed to occur with a single first-order rate constant ($k_{NO_3}$), calculated for each flight from a combination of historical ground-based VOC measurements and select VOC measurements from a PTR-MS at the UU site (see Supplemental Section S1.2 for details; Atkinson and Arey (2003)). At night, $NO_3$ serves as one of the primary tropospheric oxidants for VOCs and can react with $RO_2$ and $HO_2$ radicals to contribute to nocturnal $NO_x$ recycling (Vaughan et al., 2006). In this analysis, $NO_3$-VOC reactions were lumped and treated as a net $NO_x$ sink with values of the first order loss rate constant, $k_{NO_3}$, ranging from $1.5\times10^{-3}$ - $9.5\times10^{-3}$ $s^{-1}$ ($NO_3$ lifetime ~100 – 1000 s). These rate constants are slightly larger than average values measured during the WINTER campaign ($1.3\times10^{-4}$ to $4.6\times10^{-4}$ $s^{-1}$) (McDuffie et al., 2018b) and within the range previously reported ($3\times10^{-5}$ to $1\times10^{-2}$ $s^{-1}$) during winter 2012 at a ground site in Colorado (Wagner et al., 2013). Additional $NO_x$-regeneration from reactions of $NO_3$ with $HO_2$ and $RO_2$ radicals were not included in this analysis due to a lack of radical measurements. An under-prediction in $k_{NO_3}$ from these uncertainties would cause both an over-prediction in the loss rate constant of $N_2O_5$, as well as the subsequent production of nitrate. While uncertainties in $k_{NO_3}$ can lead to large model uncertainties during summertime conditions (e.g. Phillips et al., 2016), $NO_3$-VOC reactivity is largely reduced during the winter season as a result of lower biogenic emissions and colder temperatures that favor $N_2O_5$ in its equilibrium with $NO_3$. Sensitivity studies discussed below showed 0.2% changes in the median model-predicted nocturnal nitrate production rate associated with ± 50% changes in $k_{NO_3}$ (Table S4). The possibility of varying VOC reactivity with time was also investigated (Section S1.4.5), but resulted in a minimal (<0.1%) impact on nitrate production results presented below. The potential for other rate constants to vary with time may additionally lead to increased variability in the results presented in Section 3.3.

Third are uncertainties in assumptions regarding the start time and duration of each simulation. All simulations were initialized at 1.3 hours prior to sunset, assuming no initial concentrations of $N_2O_5$ or $ClNO_2$. The pre-sunset time of 1.3 hours was derived for the WINTER campaign, based on the time when predicted daytime $N_2O_5$ concentrations (described in Section S1.4.4 and Brown et al. (2005)) diverged from ambient observations when approaching sunset. This value was not recalculated for UWFPS simulations as daytime $N_2O_5$ calculations require measurements of $j(NO_3)$ photolysis rates, which were not measured during UWFPS. The median nocturnal nitrate production rate, however, changed by <0.3% when this pre-sunset time was varied between 0 and 2 hours. Photolysis rates during this time were also calculated from those measured during the WINTER campaign (Section S1.4.3; Shetter and Müller (1999)). While WINTER photolysis rates may have been larger than those during Utah PCAP events, the median modeled nocturnal nitrate production rate showed a small sensitivity (< 2.8%) to ± 40% changes in these values (Section S1.4.3). Additional uncertainties in air age (i.e. simulation start time and duration), however, may still serve to over-predict $N_2O_5$ loss rates and nocturnal nitrate based on previous sensitivity studies (McDuffie et al., 2018b). A combination of these assumptions will lead to a greater uncertainty in model results near sunset, as discussed in Section 3.3.2.



Fourth, air parcel mixing and deposition of gas-phase nitric acid were not included in base case simulations. Additional simulations, described in Section S1.4.2, included deposition using a first order nitric acid loss constant of $2.6\times10^{-6}$ s$^{-1}$, calculated from a boundary layer height of 800 m, deposition velocity of 2.7 cm s$^{-1}$ (Zhang et al., 2012), and gas/particle nitrate fraction of 8% from Franchin et al. (2018). The median nocturnal nitrate production rate increased by < 8% when this depositional loss of

HNO$_3$ was included. In contrast, modeled nitrate production was most sensitive (-42.2% reduction) to the addition of a 1$^{st}$ order loss processes, meant to simulate air parcel dilution and O$_3$ entrainment from vertical mixing between the RL and free troposphere (Table S4). Based on a previous analysis by Womack et al. (2019), the dilution rate constant here was estimated to be $1.3\times10^{-5}$ s$^{-1}$ in the RL, with a possible range of $1.2\text{-}2.5\times10^{-5}$ s$^{-1}$ (described in Section S1.4.1). Results from the simulations that include dilution are discussed further in the final section.

Finally, the absolute uncertainty associated with each individual nocturnal nitrate production rate was calculated from the quadrature addition of the uncertainties associated with sensitivity tests described above and in Section S1.4, as well as uncertainties in the NO$_2$, O$_3$, N$_2$O$_5$, and ClNO$_2$ measurements used as model fit parameters (< 6% for all tests). Production rates derived from model fits to observations, as well as the absolute uncertainties associated with all 17 sensitivity tests are shown as a time series in Figure S3, with dilution contributing 92% of the total uncertainty (light blue in Figure S3) on average. Both the base

case results (black dots) and those from simulations including the effects of air parcel dilution are discussed in Section 3.3.3.

## 3 Results and Discussions

### 3.1 PM$_{2.5}$ in Salt Lake Valley – Winter 2017

To provide an overview of wintertime pollution events in SLV, Figure 4 shows a time series of total PM$_{2.5}$ mass (1-hour and 24-hour averages) measured at the UDAQ Hawthorne (HW) site (Figure 1) from 1 December 2016 to 22 February 2017. Additional

time series of ground-based PM$_{2.5}$ measurements for all three Utah NAAs are provided in Franchin et al. (2018). The SLV data in Figure 4 show four pollution events that exceeded the NAAQS during the 2016-2017 winter. Calculated from 24-hour measurements, the four largest pollution events during December 2016 and January 2017 had daily PM$_{2.5}$ build-up rates that ranged from 3.7 – 15.6 μg m$^{-3}$ day$^{-1}$ (see Figure 4), encompassing the daily rates reported previously in the same valley (Whiteman et al., 2014; Silcox et al., 2012; Baasandorj et al., 2017). The last two major pollution events (10 - 22 January (Event #3) and 25 January

- 5 February (Event #4)) overlapped with the flights during UWFPS, shown by the gray shading in Figure 4. Average non-refractory (NR) PM$_1$ aerosol mass fractions measured during these periods by the TO AMS showed that the aerosol was primarily composed of NH$_4$NO$_3$ (Figure 4 pie charts). The sum of NO$_3^-$ and NH$_4^+$ contributed to 76.6% and 74.0% of the total PM$_1$ mass measured during the last two pollution episodes (74% average (Franchin et al., 2018)), which agree with previous ground-based observations (e.g. Baasandorj et al., 2017) of past seasons. Nitrate alone contributed to an average 57% and 58% of the total aerosol mass during

pollution episodes #3 and #4, respectively. During the relatively clean period sampled between 8 and 12 February 2017, the combined NH$_4^+$ + NO$_3^-$ fraction decreased to an average of 57%, with a larger relative contribution from aerosol organics. The remaining analyses here will focus on aircraft flights during the two late January pollution events (#3 and #4) to evaluate the contribution of nocturnal RL heterogeneous nitrogen chemistry to observed surface-level nitrate during pollution events.





### 3.2 Limiting and Excess Reagents for $NH_4NO_3$ Aerosol

As $NH_4NO_3$ was the principle component of $PM_{2.5}$ during pollution events in SLV (Figure 4), the contribution from heterogeneous reactive nitrogen processes is dependent on whether $NH_4NO_3$ formation is limited by the availability of gas-phase $NH_3$ or $HNO_3$. Under ambient conditions, gas-phase $NH_3$ and $HNO_3$ are assumed to be in a thermodynamic equilibrium with their particulate

equivalents ($NO_3^-(p)$ and $NH_4^+(p)$). The limiting reagent can therefore be inferred from the ratio of total oxidized ($HNO_3(g) + NO_3^-$ (p)) to total reduced nitrogen ($NH_x = NH_3(g) + NH_4^+(p)$), shown in Eq (3). This ratio does not account for other aerosol components such as $(NH4)_2SO_4$, $NH_4HSO_4$, and $NH_4Cl$, but should generally represent the $NH_4NO_3$ aerosol system when particulate concentrations of sulfate and inorganic chloride are low, as was observed during UWFPS 2017 (Figure 4 and Franchin et al. (2018)). A nitrogen ratio greater than 1 indicates that oxidized nitrogen is in excess and $NH_4NO_3$ particle formation is limited by

the presence of $NH_3$. Conversely, a ratio smaller than 1 indicates that formation is limited by the presence of $HNO_3$, which itself is limited by the oxidation rate of $NO_x$. In a $HNO_3$-limited system, $NH_4NO_3$ formation will be sensitive to changes in $HNO_3$ concentrations resulting from both day and nighttime $NO_x$ oxidation processes. Daytime $NO_x$ oxidation rates during winter will depend on specific conditions but are generally slower, such that nighttime oxidation may play a dominant role (e.g. Wood et al., 2005; Kenagy et al., 2018).

$$NRatio = \frac{HNO_3(g) + NO_3^-(p)}{NH_3(g) + NH_4^+(p)} \tag{3}$$

A time series of nitrogen ratios in SLV between 17 January and 1 February is shown in Figure 5a, calculated from 10s averaged (AMS frequency) measurements of gas and particle-phase compounds aboard the TO aircraft. Figure 5a shows that $NH_4NO_3$ particle formation in SLV during pollution episodes was largely limited by $HNO_3$ (median ratio 0.77), but highly variable (range of 0.1 -1.9) and time dependent, with the frequency of $NH_3$-limited conditions increasing throughout both late January

pollution events. The color scale in Figure 5a and the vertical profiles of average and 10-90[th] percentile nitrogen ratios in Figure 5b further show that the lowest nitrogen ratios corresponded to the lowest altitudes. These results of $HNO_3$-limitation near the ground are consistent with all previous ground-based observations that show exclusive $HNO_3$-limitation in SLV (Kelly et al., 2013; Utah Division of Air Quality, 2014b). The increased frequency of $NH_3$-limited points throughout both pollution episodes (Figure 5a), however, is opposite the trend predicted by Baasandorj et al. (2017), who suggested that observed surface-level oxidant

depletion should lead to more $HNO_3$-limited conditions over time. Events of $NH_3$-limitation (excess $HNO_3$) during 2017, however, only occurred at the highest altitudes (panel b) and their increasing frequency with time (panel a) is consistent with these events reflecting negative $NH_3$ gradients away from the surface and/or the production of $HNO_3$ aloft from nocturnal $N_2O_5$ chemistry. The rate of $HNO_3$ production from nocturnal heterogeneous chemistry is expected to be maximized at higher altitudes, removed from NO emissions and $O_3$ titration at the surface (Figure 2). Results here are also consistent with aerosol thermodynamic modeling

studies by Franchin et al. (2018) who found that simulations of total $PM_1$ mass during UWFPS flights over SLV were proportionally sensitive to 50% reductions in total nitrate. Additional simulations, however, also resulted in near 50% $PM_1$ reductions with 50% reductions in total ammonium ($NH_3+NH_4^+$) (Franchin et al., 2018), indicating that 50% ammonium reductions may be enough to shift SLV from the $HNO_3$ to $NH_3$-limited regime, consistent with nitrogen ratios in Figure 5 approaching and exceeding 1.





### 3.3 Nitrate Production via Heterogeneous Reactive Nitrogen Chemistry

The absolute amount of nitrate chemically produced from heterogeneous chemistry will depend on the production rate of the nitrate radical and gas-phase $N_2O_5$ (Section 3.3.1), $N_2O_5$ aerosol uptake efficiency (Section 3.3.2), and yields of $ClNO_2$ and $HNO_3$ (Section 3.3.2), which are quantified below. The final section (Section 3.3.3) presents forward-integrated box model simulations to further

quantify the nocturnal nitrate production rate and estimate the contribution of this chemistry to $NH_4NO_3$ formation during January 2017 in SLV.

### 3.3.1 Maximum Instantaneous Nitrate Production Rates

An upper limit estimate of the instantaneous rate of aerosol nitrate production from heterogeneous $N_2O_5$ chemistry, $P_{NO_3^-}$, can be calculated as two times the gas-phase $N_2O_5$ production rate, $P_{N_2O_5}$. These instantaneous production rates are calculated from the

rate limiting reaction between $NO_2$ and $O_3$, which forms the nitrate radical (Eqs. (4) – (6)). In Eq. (4), $P_{N_2O_5}$ is calculated in units of molec. $cm^{-3}$ $s^{-1}$ but is typically reported in units of ppbv $hr^{-1}$ after conversion using the ambient air concentration ($ND$) and conversion factors for seconds to hours (3600) and mixing ratio to ppbv ($1 \times 10^9$). The reaction kinetics in Eq. (5) between $NO_2$ and $O_3$ are from the 2008 IUPAC recommendation (IUPAC, 2008). In Eq. (6), $P_{NO_3^-}$ is then calculated as 2 times $P_{N_2O_5}$ after it has been converted to units of $\mu g$ $m^{-3}$ $hr^{-1}$, as detailed in Supplemental Section S2. This calculation assumes: 1) $N_2O_5$ is produced

quantitatively from $NO_3$ (i.e. no competing reaction of $NO_3$ + VOC), 2) $N_2O_5$ is produced at the rate of $NO_3$ production (valid under cold conditions that shift the $NO_3$-$N_2O_5$ equilibrium to favor of $N_2O_5$), 3) $N_2O_5$ is efficiently taken up onto aerosol, and 4) aqueous-phase reactions form two molecules of $HNO_3$ for every molecule of $N_2O_5$ (i.e. $\phi(ClNO_2) = 0$).

$$P_{N_2O_5} \text{ [ppbv hr}^{-1}] = \frac{k_4[O_3][NO_2]}{ND \text{ [molec. cm}^{-3}]} * 3600 \text{ [s hr}^{-1}] * 1 \times 10^9 \text{ [ppbv]} \tag{4}$$

$$k_4 \text{ [cm}^3 \text{ molecule}^{-1} \text{ s}^{-1}] = 1.4 \times 10^{-13} e^{(-2470/T)} \tag{5}$$

$$P_{NO_3^-} [\mu g \text{ m}^{-3} \text{ hr}^{-1}] = 2 * (P_{N_2O_5} [\mu g \text{ m}^{-3} \text{ hr}^{-1}]) \tag{6}$$

The value of $P_{NO_3^-}$ is expected to vary with altitude due to boundary layer dynamics and surface $NO_x$ emissions that can

deplete $O_3$ at night near the surface, as described previously in Baasandorj et al. (2017). The time series in Figure 6a illustrates that the $O_3$ measured at HW was frequently absent at night during the 3rd and 4th pollution events in January 2017. As surface-level $O_3$ was titrated overnight, ground-site data cannot provide direct information about $P_{NO_3^-}$ aloft in the RL. In the absence of vertical observations during pollution events in 2016, a previous analysis by Baasandorj et al. (2017) used late afternoon measurements at the HW ground site to predict $N_2O_5$ production rates ($P_{N_2O_5}$) in the RL that varied from 0 up to ~ 2 ppbv $hr^{-1}$ (~ $0 - 5$ $\mu g$ $m^{-3}$ $hr^{-1}$),

but with values frequently < 1 ppbv $hr^{-1}$. These values correspond to instantaneous nitrate production rates ($P_{NO_3^-}$) of ~ $0 - 10$ $\mu g$ $m^{-3}$ $hr^{-1}$, with typical values closer to 5 $\mu g$ $m^{-3}$ $hr^{-1}$. Late afternoon estimates from the same site during 2017 (dashed lines in Figure 6, from Eq. (6)), suggest smaller $P_{NO_3^-}$ rates in 2017 than in 2016, with values between 1 and 5 $\mu g$ $m^{-3}$ $hr^{-1}$ during UWFPS pollution events (Figure 6a).

The bottom panels of Figure 6b show the binned, vertical profiles of median, 25th, and 75th percentile instantaneous $P_{NO_3^-}$

values, along with aircraft observations of $O_3$, $NO_2$, and $PM_1$ for all UWFPS night flights (red shaded regions in Figure 6a). The vertical profiles show a relatively uniform distribution of $P_{NO_3^-}$ with altitude through the lowest 600 m. The dashed black lines





also show that the number of points in each altitude bin was weighted toward the 100-500 m altitude range. The median instantaneous $P_{NO_3^-}$ value in this polluted layer (0-600 mAGL) was 1.6 µg m$^{-3}$ hr$^{-1}$ (N = 21666). This value is at the low range of estimates of 1.6 - 5 µg m$^{-3}$ hr$^{-1}$ that are predicted from late afternoon ground-based observations on each flight day (dashed line in the middle panel of Figure 6a), following the method of Baasandorj et al. (2017).

Vertical profiles in Figure 6b do not show evidence for a reduction in $P_{NO_3^-}$ or O$_3$ near the surface, as is expected for O$_3$ titration near the ground level (shown in panel a). The distribution in panel b, however, is affected by the location of the missed approaches / landings in the SLV (Salt Lake International and South Valley Regional airfields), which are further from the urban center of Salt Lake City than the HW ground site (see Figure 1). Vertical profiles to the surface over urban Salt Lake City were not possible due to a lack of airfields for missed approaches. Instead, the SLV flights often executed box patterns over the eastern

Salt Lake basin at several altitudes. Figure 7 shows the vertical distribution of $P_{NO_3^-}$ values from these boxes on January 28 - 29 between 21:20 – 00:30 local time, compared to $P_{NO_3^-}$ measured at surface level during the same interval. At 300 and 500 m AGL, the median (and interquartile range) $P_{NO_3^-}$ was 2.2 (2.1 to 2.4) and 1.9 (1.8 to 2.1) µg m$^{-3}$ hr$^{-1}$, while at 650 m, slightly above the most concentrated pollution layer, it was 0.5 (0.3 to 0.7) µg m$^{-3}$ hr$^{-1}$. The median value at the HW ground site, directly below the aircraft, was 0.02 (0.01 to 0.2) µg m$^{-3}$. These plots demonstrate that $P_{NO_3^-}$ is typically low or zero at night near the surface within

the urban area of Salt Lake City, but large within the RL. Away from the urban area, the vertical distributions of $P_{NO_3^-}$ are also likely more uniform (Figure 6b) due to the lack of O$_3$ titration within the nocturnal boundary layer. In the final section below, nightly integration of these instantaneous $P_{NO_3^-}$ values are compared to box model predictions of total nitrate.

### 3.3.2 Modeled Uptake Coefficients and Production Yields

Both the aerosol uptake efficiency of N$_2$O$_5$ ($\gamma$(N$_2$O$_5$)) and the production yield of ClNO$_2$ ($\phi$(ClNO$_2$)) are highly variable, dependent

on aerosol composition, and can impact the absolute amount of nitrate formed from nocturnal heterogeneous nitrogen chemistry. The nighttime formation of nitrate, however, is only limited by these processes when N$_2$O$_5$ uptake is inefficient and is instead limited by the oxidation rate of NO$_2$ (R1) (discussed above) at sufficiently large values of $\gamma$(N$_2$O$_5$).

As described in Section 2.2, an iterative box model was fit to observations of NO$_2$, O$_3$, N$_2$O$_5$, and ClNO$_2$ to quantify $\gamma$(N$_2$O$_5$) and $\phi$(ClNO$_2$) during pollution events. For SLV alone (N = 1030), the distribution in Figure 8 shows that $\gamma$(N$_2$O$_5$) values

ranged four orders of magnitude from 1 ×10$^{-3}$ to > 1. Values approaching or exceeding 1 are unphysical and suggest artifacts in the $\gamma$(N$_2$O$_5$) determinations for UWFPS (see below), at least for the largest values. Values of $\phi$(ClNO$_2$) encompassed the entire possible range of 0 to 1 (Figure 8). The medians for this subset were 0.076 and 0.220 for $\gamma$(N$_2$O$_5$) and $\phi$(ClNO$_2$), respectively. For all UWFPS flights between 16 January and 1 February 2017, the median $\gamma$(N$_2$O$_5$) and $\phi$(ClNO$_2$) values in the RL (N = 2195) were 0.049 and 0.256, respectively, derived from box-model fits to observations. These values are compared to multiple derivation

methods further below.

Compared to previous studies, the median $\gamma$(N$_2$O$_5$) over SLV was twice as large as the mode derived with a similar model using data from the Nitrogen, Aerosol Composition, and Halogens on a Tall Tower (NACHTT) campaign near Denver, Colorado in winter 2011 (Wagner et al., 2013). Similarly, the median was over 5 times larger than the median calculated using the same model from the 2015 WINTER campaign (McDuffie et al., 2018b). The largest values during UWFPS exceeded those from both

WINTER and NACHTT studies, while the smallest values were also larger than either of the respective minimums. The two most common suppression mechanisms that lead to reductions in $\gamma$(N$_2$O$_5$) are associated with the presence of organic material and nitrate



in the aerosol phase. Insoluble aerosol organics have been shown to suppress $N_2O_5$ uptake in previous laboratory studies (e.g. Griffiths et al., 2009; Thornton et al., 2003; McNeill et al., 2006; Thornton and Abbatt, 2005; Cosman et al., 2008; Badger et al., 2006; Folkers et al., 2003) and large organic mass fractions have been associated with $\gamma(N_2O_5)$ reductions in past field studies (Bertram et al., 2009; McDuffie et al., 2018b). The average dry mass fraction of aerosol organics (i.e. organic mass / total dry

aerosol mass) during SLV pollution events was less than half of that observed during the WINTER campaign (~18% vs 40%) and 40% lower than the average during NACHTT (27%, (Wagner et al., 2013)). Aerosol nitrate can also suppress uptake as soluble nitrate facilitates the reformation of gas-phase $N_2O_5$ (Bertram and Thornton, 2009; Griffiths et al., 2009), and nitrate mass fractions have been negativly correlated with $\gamma(N_2O_5)$ in previous field-studies (Wagner et al., 2013; Morgan et al., 2015; Riedel et al., 2012; Bertram et al., 2009; McDuffie et al., 2018b). The presence of sufficient aerosol water, however, can offset this nitrate suppression

by promoting $N_2O_5$ aqueous solvation and reaction (e.g. Bertram and Thornton, 2009; Griffiths et al., 2009; Mentel et al., 1999; Wahner et al., 1998), resulting in increases in $\gamma(N_2O_5)$ with the ratio of water to nitrate (McDuffie et al., 2018b). The average dry mass fraction of aerosol nitrate was much larger during UWFPS (60%) than during NACHTT (30%, Wagner et al. (2013)) or WINTER (15%, McDuffie et al. (2018b)). High humidity conditions during UWFPS (77% average RH during pollution events) resulted in average aerosol water mass fractions (i.e. water mass / aerosol dry mass + water mass) near 70%, as calculated with an

aerosol thermodynamic model, described in Franchin et al. (2018). This higher RH likely contributed to efficient $N_2O_5$ uptake during UWFPS despite the presence of aerosol nitrate. In fact, the largest 25% of UWFPS $\gamma(N_2O_5)$ values exceed the largest value (0.175) that has been reported from recent field studies (Figure 4 in McDuffie et al. (2018b)).

       The median $\phi(ClNO_2)$ value of 0.220 during SLV pollution events was 4 times larger than during the NACHTT campaign (Riedel et al., 2013; Wagner et al., 2013), but within a factor of 2 larger than the median derived during WINTER over the U.S.

east coast (McDuffie et al., 2018a). The SLV median was also similar to medians reported from previous ground-based studies across North America (Mielke et al., 2016; Mielke et al., 2011; Mielke et al., 2013; Wagner et al., 2012; Thornton et al., 2010). Heterogeneous $ClNO_2$ production requires aerosol chloride (R6) (e.g. Behnke et al., 1997) and though a consistent geographic pattern in $\phi(ClNO_2)$ has not emerged from past studies (Figure 2 in McDuffie et al. (2018a)), heterogeneous chemistry in the vicinity of the Great Salt Lake appears to produce $ClNO_2$ with the same efficiency as comparable measurements near North

American ocean coastlines. $ClNO_2$ production yields, however, remain smaller than those predicted based on measured aerosol composition, as discussed below.

       While large $\gamma(N_2O_5)$ and moderate $\phi(ClNO_2)$ values indicate efficient nitrate production from heterogeneous chemistry during UWFPS, these values may be upper and lower limits, respectively. As discussed in Section 2.1, limited observations of VOC and photolysis rates, as well as uncertainties in air age, and dilution may cause the $k_{N_2O_5}$ and $k_{ClNO_2}$ values (and subsequent

$\gamma(N_2O_5)$ and $\phi(ClNO_2)$) to be over- and under-predicted, respectively. This is more likely near sunset where the model has an increased sensitivity to assumptions in simulation start time (McDuffie et al., 2018b). Uncertainties in gas-phase measurements may also contribute to uncertainties in the model predictions, though the level of uncertainty associated with these parameters is small (Table S4). As summarized in Table S4, the box model is not highly sensitive to most sources of uncertainty, and the model-derived values of $k_{N_2O_5}$ are consistent with those derived from observations (discussed below).

Independent of the model fits of $k_{N_2O_5}$ and $k_{ClNO_2}$, unphysically large $\gamma(N_2O_5)$ values (> 0.1 in Figure 8) may alternatively be an artifact arising from under-measurement of ambient aerosol SA. Low aerosol SA would bias high the $\gamma(N_2O_5)$ calculation in Eq. (1) without influencing the model derivations of $k_{N_2O_5}$ and $k_{ClNO_2}$. In this study, wet aerosol SA was calculated as described




above by applying a relative humidity-dependent growth factor curve to the measured dry $PM_1$ SA. Despite large concentrations of total dry SA (Figure S2), an under-prediction in the wet SA could arise from uncertainties in the hygroscopic growth curve or additional unmeasured SA from large particles (> 1 μm). Both factors would be exacerbated by the high humidity conditions encountered during UWFPS since large, hydrated particles would not be sampled efficiently by the aerosol inlet and hygroscopic

growth curves are highly uncertain above ~ 95% RH (corresponding to 6.7% of the SLV data). A third possible cause of under-measured SA is the presence of fog under high humidity conditions. Fog is well known to promote rapid heterogeneous processes (Lelieveld and Crutzen, 1990), and is associated with surface areas that can be orders of magnitude larger than accumulation mode aerosol. For example, fog has been demonstrated to lead to rapid $N_2O_5$ loss at a ground site in Hong Kong, during November - December 2013 (Brown et al., 2016). It is therefore possible that unmeasured SA under high humidity conditions could bias the

calculated $\gamma(N_2O_5)$ values high relative to values reported in previous literature. Any bias caused by aerosol SA, however, would not impact the model-derived $k_{N_2O_5}$ and $k_{ClNO_2}$ values that are used in the final analysis below.

        To further evaluate the UWFPS $\gamma(N_2O_5)$ and $\phi(ClNO_2)$ values, box model determinations are compared to two other derivation methods in Figures 8 and S5. The first method calculates $\gamma(N_2O_5)$ from $NO_2$, $O_3$, and $N_2O_5$ observations, based on the steady state approximation ($\gamma(N_2O_5)_{ss}$), described by Brown et al. (2003) and defined in Supplemental Section S4.1. This method

shows excellent agreement with box model results (Figure 8 and S5). The steady state method has been shown in previous analyses to over-predict $\gamma(N_2O_5)$ values under cold, high $NO_x$ conditions, but only if the first order rate constants for $NO_3$ and $N_2O_5$ loss ($k_{NO_3}$ and $k_{N_2O_5}$) are modest (Brown et al., 2003). Both the steady state and box model $\gamma(N_2O_5)$ values are consistent with a rapid first order loss constant of $N_2O_5$ (median $k_{N_2O_5}$ = $1.1\times10^{-3}$ $s^{-1}$), suggesting the steady state approach is valid for SLV conditions. The corresponding median lifetime ($1/k_{N_2O_5}$) of 14 minutes is, for example, much shorter than the lifetimes of 2-18 hours calculated

from a previous steady state analysis of aircraft measurements over Texas in fall 2006 (Brown et al., 2009). Nevertheless, the color scale in Figure S5 shows that the largest $\gamma(N_2O_5)$ values ($\geq 0.1$) were exclusively derived for air sampled within 3 hours of sunset (4.3 hr simulation time), where previous analysis has shown the steady state approximation to be least reliable. As Figure S5 shows large $\gamma(N_2O_5)$ determinations from both the box model and the steady state analysis during this time, there may be a common bias between the methods if these values are indeed too large.

25        The second method calculates both $\gamma(N_2O_5)$ and $\phi(ClNO_2)$ using laboratory-based parameterizations by Bertram and Thornton (2009) (BT09), based on aerosol volume-to-surface area ratio, $N_2O_5$ solubility (Fried et al., 1994), aerosol molarities of water, nitrate, and chloride (calculated as described in Section S4.2), and laboratory-derived reaction rate constant ratios. Further details of each parameterization are provided in Supplemental Section S4.2. These parameterizations have had mixed success in reproducing previous field-derived values (e.g. Bertram et al., 2009; Riedel et al., 2012; McDuffie et al., 2018b; McDuffie et al.,

2018a), but are commonly used to predict $N_2O_5$ uptake and $ClNO_2$ production on internally-mixed inorganic aerosol when $N_2O_5$ chemistry is included in global models (e.g. Sarwar et al., 2014; Shah et al., 2018; Wang et al., 2018).

        Results in Figure S5 show that the median $\gamma(N_2O_5)$ value predicted by the BT09 parameterization is within a factor of 2 of the box model median, but that this parameterization does not reproduce the observed variability (Figures 8). For $\phi(ClNO_2)$, the BT09 parameterization largely over-predicts model-derived values with a median of 0.66 relative to the model median of 0.22

(Figure S5). This over-prediction is consistent with all previous studies to compare parameterized and field-derived $\phi(ClNO_2)$ results (Wagner et al., 2013; Wang et al., 2017b; Ryder et al., 2015; Thornton et al., 2010; Riedel et al., 2013; Wang et al., 2017a; Tham et al., 2018; McDuffie et al., 2018a). These results also suggest that the parameterization would need to be reduced by 68%




for agreement with the box model median, similar to the 74-85% reduction required for agreement of this parameterization with the WINTER campaign median (McDuffie et al., 2018a). The possible presence of additional, refractory-phase chloride (i.e. NaCl, CaCl$_2$, and KCl) in the accumulation mode would increase the predicted $\gamma(N_2O_5)$ and improve agreement with the box model, but would further degrade the agreement of $\phi(ClNO_2)$.

Lastly, the empirically-based $\gamma(N_2O_5)$ parameterization from McDuffie et al. (2018b) was applied to UWFPS data, though only an estimated range for the campaign median is presented here due to uncertainties in the aerosol O:C ratio and aerosol organic density, both required for this calculation (discussed in Section S4.2). This parameterization models $N_2O_5$ uptake onto an aqueous inorganic particle with a resistive organic coating, with a thickness determined by the volume ratio of inorganic to total aerosol components (McDuffie et al., 2018b; Riemer et al., 2009; Anttila et al., 2006). By estimating a range of O:C ratios using the

improved-ambient O:C ratio method from Canagaratna et al. (2015) and AMS organic $m/z$ 44 fraction (Figure 6, Franchin et al. (2018)), assuming an organic density of 1.3 g/cm$^3$ (e.g. Kuwata et al., 2012) to estimate the organic-associated volume, and applying additional constants described in Section S4.2, this parameterization estimated a median $\gamma(N_2O_5)$ between 60 and 85% lower than the box model. Though there are large uncertainties in the required parameters, these results suggest that during pollution events: 1) aerosol organics are not surface active, 2) aerosol organics are not resistive toward $N_2O_5$, or 3) box model $\gamma(N_2O_5)$ values

are over-predicted due to missing SA (e.g. fog, Section 3.3.2) or other simplifying assumptions (e.g. dilution) discussed above.

         Despite disagreement between the box model and parameterizations, the $\gamma(N_2O_5)$ values predicted by all three methods are large enough, in combination with the large measured aerosol SA, to fall within the range where models of nighttime chemistry are insensitive to variation in uptake efficiency (e.g. Macintyre and Evans, 2010; Riemer et al., 2003). Under these conditions, the $NO_2$ gas-phase oxidation rate (i.e. $P_{NO_3^-}$) becomes the limiting factor to $HNO_3$ formation relative to $N_2O_5$ uptake. As further

evidence, the median lifetime of $NO_2$ with respect to $O_3$ ($\tau_{NO_2} = 1/(k_1[O_3])$) was 9 hours during pollution events in SLV, while the equivalent lifetime of $N_2O_5$ ($\tau_{N_2O_5} = 1/k_{N_2O_5}$) was 14 minutes. Further to this point, explicit box modeling of day and nighttime chemical processes during UWFPS by Womack et al. (2019) showed that the production of $O_{x,total}$ (= $NO_2 + O_3 + 2*NO_3 + 1.5*(HNO_3 + particulate\ nitrate) + ClNO_2 + 3*N_2O_5 + others$) was insensitive (<1.5%) to order-of-magnitude changes in $\gamma(N_2O_5)$. Short lifetimes of $N_2O_5$ relative to $NO_2$, as well as nitrate insensitivity to $\gamma(N_2O_5)$, both indicate that nocturnal heterogeneous

chemistry contributes to $NH_4NO_3$ formation, but that absolute production is limited by gas-phase kinetics rather than aerosol composition and $\gamma(N_2O_5)$. This insensitivity to $\gamma(N_2O_5)$ provides confidence in the ability of the box model to predict the magnitude of nocturnal nitrate production in SLV, regardless of uncertainties in $\gamma(N_2O_5)$.

### 3.3.3 Modeled Nocturnal Nitrate Production Rates and Contribution of Heterogeneous Chemistry to Total NH$_4$NO$_3$ Aerosol Accumulation Rates

As described in Section 2.2 and shown in Figure 3, the box model simulates the amount of total nitrate ($HNO_3 + NO_3^-$) produced from heterogeneous chemistry over the course of a single night. This amount of nitrate, in units of $\mu g\ m^{-3}\ night^{-1}$, is in addition to any nitrate present at sunset from the previous day (e.g. Figure 3). Figure 9 shows the distribution of nightly nitrate production predicted by base case simulations (N = 1033), with a median of 9.9 $\mu g\ m^{-3}$ nitrate night$^{-1}$.

         Comparisons between the base case results and integrated $P_{NO_3^-}$ from Section 3.3.1 also suggest that nocturnal nitrate

production is limited by the rate of $NO_2$ oxidation rather than the efficiency of $N_2O_5$ aerosol uptake. Based on the calculations in



Section 3.1, upper-limit $P_{NO_3^-}$ values, integrated over an average 14 hour night and reduced to account for a $\phi(ClNO_2)$ value of 0.2, ranged from < 0.5 to > 40 µg m$^{-3}$ night$^{-1}$, with a median of 20.2 µg m$^{-3}$ night$^{-1}$ (N = 21666). To more directly compare with box model results, the subset of points with simultaneous $\gamma(N_2O_5)$ determinations had a median of 10.6 µg m$^{-3}$ night$^{-1}$, which is slightly larger, but agrees well with the box-model predicted median of 9.9 µg m$^{-3}$ night$^{-1}$. As described in Section 3.3.1, the $P_{NO_3^-}$

calculation assumes efficient $N_2O_5$ uptake and only considers nitrate production to be limited by gas-phase kinetics. Observed agreement between the integrated $P_{NO_3^-}$ values and box model-predicted production rates, therefore suggests that nitrate production may be largely limited by gas-phase oxidation rather than multi-phase processes.

Uncertainties associated with base case production rates are discussed in Section 2.2.2 and shown as a time series in Figure S3. Air parcel dilution associated with vertical mixing was the largest source of uncertainty (Table S4, Figure S3). This

process was not included in base case simulations, though mixing / dilution has been observed and predicted in an analysis of WINTER nighttime flights (Kenagy et al., 2018; McDuffie et al., 2018b). Estimating the impact of dilution by including a single first order dilution rate constant ($k_{dilution}$) of $1.3 \times 10^{-5}$ s$^{-1}$ reduced the median nocturnal nitrate production rate by 42% to 5.7 µg m$^{-3}$ night$^{-1}$, shown in comparison to base case simulations in Figure 9. This $k_{dilution}$ rate constant was derived, as described in Womack et al. (2019), by fitting a box model to best reproduce the build-up of $O_{x,total}$ observed between 28 January and 1 February

at the UU ground site in SLV. Following Womack et al. (2019), this entrainment rate constant of $8 \times 10^{-6}$ s$^{-1}$ was then scaled up by 40% to represent the reduced volume of the nocturnal RL relative to the mixed daytime boundary layer, for which this rate constant was derived. The resulting $k_{dilution}$ of $1.3 \times 10^{-5}$ s$^{-1}$ is ~60% lower than $k_{dilution}$ from the WINTER campaign, derived from observations of NO$_y$ (= NO + NO$_2$ + NO$_3$ + 2*N$_2$O$_5$ + ClNO$_2$ + RONO$_2$...) overnight in a single RL air parcel over the eastern U.S. coast (McDuffie et al., 2018b). As processes relevant to RL dilution were not directly measured during UWFPS, there are

uncertainties associated with this $k_{dilution}$ estimation. For instance, based on the modeled surface albedo in Womack et al. (2019), $k_{dilution}$ could have reproduced observed $O_{x,total}$ mixing ratios with scaled values ranging between $1.2 \times 10^{-5}$ and $2.5 \times 10^{-5}$ s$^{-1}$ (Figure S10, Womack et al., 2019). This particular range of loss rate constants predicts median nitrate production rates in SLV between 3.6 and 6.1 µg m$^{-3}$ night$^{-1}$.

Modeled nitrate production rates are further compared in Figure 10 to the average daily accumulation of surface-level

nitrate aerosol during pollution event #4 at the HW ground site. This ground-based accumulation rate (red diamond in Figure 10a) was taken as the slope of the 24-hr average PM$_{2.5}$ observations at HW (scaled by 0.58; average NO$_3^-$ fraction from Figure 4) during the first six days of event #4, before it began to degrade on 1 February 2017 (Figure 10b). Only data from event #4 data are here assessed as this was the only PCAP sampled with the aircraft on multiple nights. Figure 10a shows this average, 24-hour surface accumulation rate of 4.6 µg m$^{-3}$ day$^{-1}$ (red diamond) compared to the 10$^{th}$ – 90$^{th}$ percentile distributions, medians, and averages of

the nocturnal production rates predicted by base case box model simulations (gray) and simulations including the effects of 24-hour dilution (blue), described below.

Comparing modeled RL chemical nitrate production to the observed ground-based accumulation rate can provide an estimate for the $N_2O_5$ uptake contribution to total particulate nitrate production in SLV. Direct comparison is difficult, however, as the 24-hour ground-based accumulation rate includes contributions from night- and daytime chemical production, but also

depends on dilution and mixing processes. For example, the amount of nocturnally produced nitrate observed at the surface will depend on mixing of nitrate aerosol to the surface from the RL during morning boundary layer expansion (Figure 2). As a result, base case box model predictions (no dilution consideration) in Figure 10a (gray) had a median of 8.6 µg m$^{-3}$ night$^{-1}$, nearly twice





as large as the observed, 24-hour average ground-based accumulation rate. Therefore, to more directly compare box model predictions and ground-based observations, Figure 10a also shows the results from simulations that included loss from both nocturnal and daytime dilution. At night, $k_{dilution}$ values of $1.2 \times 10^{-5}$ s$^{-1}$ (L), $1.3 \times 10^{-5}$ s$^{-1}$ (M), and $2.5 \times 10^{-5}$ s$^{-1}$ (H) (blue) were applied to all modeled species as described above. Modeled nighttime nitrate (e.g. Figure 9) was then further diluted for ~10 hours

(24 − 14 night) at 60% of the nocturnal dilution rate ($k_{dilution}*0.6 = 8 \times 10^{-6}$ s$^{-1}$) to reflect the increased volume of the daytime boundary layer, following Womack et al. (2019). For a single 24-hour period, this resulted in a net median of 3.9 μg m$^{-3}$ nitrate produced from nocturnal heterogeneous N$_2$O$_5$ uptake, with a range of medians between 1.9 and 4.2 μg m$^{-3}$ day$^{-1}$ when considering the extended range of dilution rate constants from Womack et al. (2019).

Comparison of modeled rates to the observed, daily surface build-up of 4.6 μg m$^{-3}$ day$^{-1}$, suggests that on average, nitrate

produced from heterogenous chemistry can generally account for the nitrate accumulation observed at the surface. This result is qualitatively consistent with an observational analysis by Pusede et al. (2016), who determined that nocturnal heterogeneous chemistry was the main source of regional aerosol nitrate during wintertime pollution events in the San Joaquin Valley. A box model analysis of this same event by Womack et al. (2019), however, also showed that photochemical nitrate production is also occurring during these events with roughly equal contributions between photochemical and nocturnal nitrate production pathways.

Therefore, while results in Figure 10a (including dilution) predict a median nocturnal fractional contribution of 86% (ranging between 42 and 91%), confirmation and further quantification of this result will require additional, vertically resolved measurements of aerosol composition, gas-phase precursors, and physical parameters, as well as more sophisticated modeling of these multi-day pollution accumulation events with 3D-chemical transport models.

## 4 Summary and Conclusions

Aerosol and gas-phase measurements collected during the 2017 UWFPS campaign showed multiple pollution events that exceeded PM$_{2.5}$ standards in SLV, the most populated region in the state of Utah. During these events, aerosol particles were largely composed of NH$_4$NO$_3$, which forms from the reaction between gas-phase NH$_3$ and HNO$_3$. While NH$_3$ is emitted from surface sources, HNO$_3$ is chemically formed from the oxidation of NO$_x$ emissions. This oxidation can occur through daytime reactions with the photochemical OH radical, or nocturnal heterogeneous reactions involving NO$_3$ and N$_2$O$_5$. The contribution of nocturnal

chemistry to PM$_{2.5}$ formation in SLV is dependent on whether NH$_4$NO$_3$ formation is NH$_3$- or HNO$_3$-limited, as well as the NO$_3$ production rate, N$_2$O$_5$ uptake efficiency, ClNO$_2$ and HNO$_3$ production yields, and loss processes such as air parcel dilution.

Vertically resolved measurements of gas and particulate phase oxidized and reduced nitrogen in SLV showed that NH$_4$NO$_3$ formation during pollution events was nearly always HNO$_3$ limited, but that oxidized and reduced nitrogen approached equivalence as pollution events progressed. This reagent balance analysis is consistent with aerosol thermodynamic modeling

presented in Franchin et al. (2018), which predicted that all three major valleys in Wasatch region were sensitive to nitrate reductions, and that SLV was also sensitive to NH$_3$ reductions. Both observation and modeling-based analyses agreed that NH$_4$NO$_3$ formation in the RL was largely HNO$_3$-limited during pollution events, providing the possibility of a large contribution from nocturnal heterogeneous chemistry to HNO$_3$ and PM$_{2.5}$ mass.

Analysis of vertically-resolved, calculated nitrate production rates (an upper-limit estimate due to heterogeneous HNO$_3$

formation, $P_{NO_3^-}$) and results from an observationally-constrained chemical box model, suggest that nocturnal chemistry is an efficient mechanism for PM$_{2.5}$ production in SLV during pollution events. Nitrate production rates had a median of 1.6 μg m$^{-3}$ hr$^{-}$



[1], while values of $\gamma(N_2O_5)$ and $\phi(ClNO_2)$ had medians of 0.076 and 0.220, respectively, during pollution events. Values of $\gamma(N_2O_5)$ were larger than previous field-based determinations (e.g. McDuffie et al., 2018b) and those predicted from the Bertram and Thornton (2009) parameterization, but were in agreement with values derived using the $N_2O_5$ steady state approach. The median $\phi(ClNO_2)$ value was larger than that derived from aircraft observations over the eastern US coast, but were simultaneously

overpredicted by 68% by the Bertram and Thornton (2009) parameterization, which uses measurements of aerosol chloride and water estimations.

While the box model has uncertainties associated with limited available measurements and model assumptions, the large measured aerosol SA, efficient $N_2O_5$ uptake coefficients, and moderate $ClNO_2$ yields resulted in nightly modeled nitrate production rates that were largely insensitive to specific values of derived parameters. Agreement between base case modeled nightly nitrate

production (9.9 $\mu$g m$^{-3}$ night$^{-1}$) and that calculated from $P_{NO_3}-$ values (10.6 $\mu$g m$^{-3}$ night$^{-1}$) alternatively suggests that nitrate production is more sensitive to gas-phase $NO_2$ oxidation rates than $\gamma(N_2O_5)$, providing confidence in the model's predictions of nocturnal nitrate. Of the parameters tested, the model was most sensitive to loss through air parcel dilution, with a 42% reduction to 5.2 $\mu$g m$^{-3}$ nitrate night$^{-1}$ when including a nocturnal $k_{dilution}$ rate constant of $1.3\times10^{-5}$ s$^{-1}$. When considering the possible effects of 24-hour dilution, model simulations predicted a reduced median of 3.9 $\mu$g m$^{-3}$ nitrate day$^{-1}$, corresponding to 86% (median) of

the net aerosol nitrate accumulation that was observed at a SLV ground site. Due to model uncertainties and sensitivities to dilution, further quantification of this result will require additional vertically-resolved measurements and photochemical / 3D modeling analyses. These results however, highlight the importance of nocturnal chemistry in the formation of PM$_{2.5}$ in SLV and can provide constraints for regulatory models of PM$_{2.5}$, used to assess control strategies in this populated non-attainment area.

**Author Contributions**

During the UWFPS campaign, EEM, CCW, DLF, and WPD were responsible for the CRD gas-phase measurements, AF and AM for the AMS particle measurements, LG, BHL, and JAT for the I$^-$TOF-CIMS measurements, and AM and JM for the NH$_3$ instrument. MB and SSB organized the UWFPS campaign with technical support from WPD. EEM developed the box model code and preformed the analyses with support from CCW and SSB. EEM prepared the manuscript with contributions from co-authors.

**Acknowledgements**

The authors would like to thank NOAA Aircraft Operations, staff, and pilots deployed as part of the UWFPS campaign, Jason Clark, Rob Mitchell, and Rob Militec. NOAA acknowledges support for Twin Otter flights from the Utah Division of Air Quality under agreement number 16-049696. Data from the UWFPS campaign can be found at the NOAA website:

https://www.esrl.noaa.gov/csd/groups/csd7/ measurements/2017uwfps/. Code written in IGOR Pro for the iterative box model can be found at: https://esrl.noaa.gov/csd/groups/csd7/measurements/2015winter/pubs/. All referenced supplemental text, figures, and tables can be found in the supporting information.



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





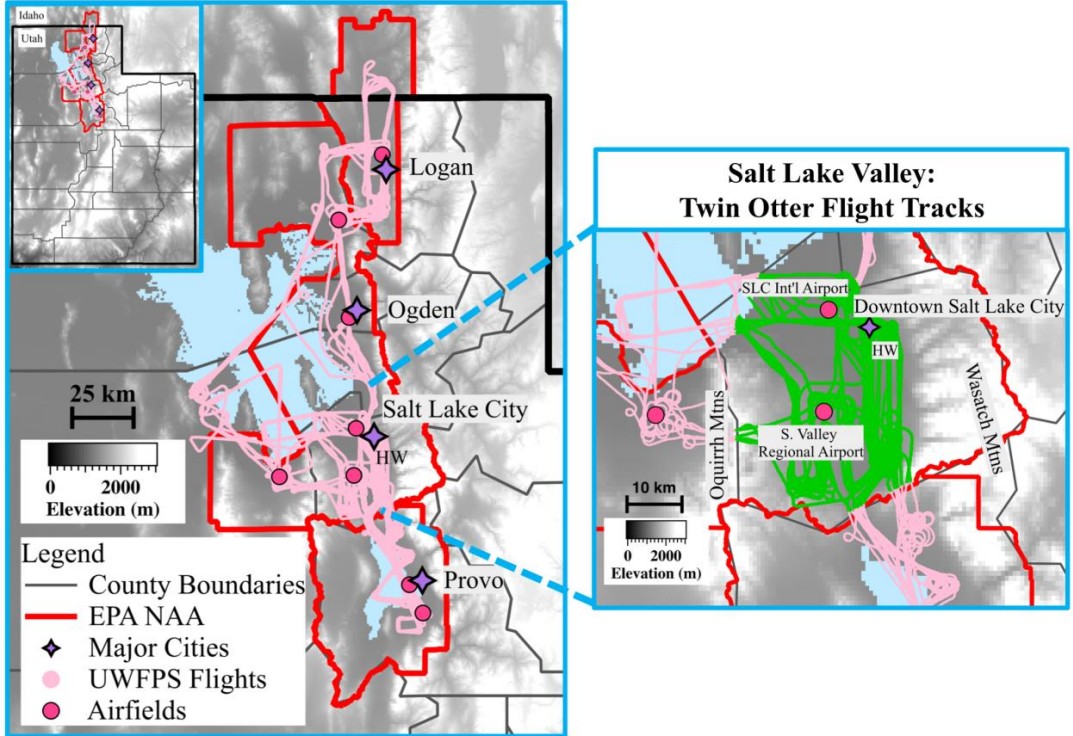

**Figure 1. (Left) Elevation Map of Utah's Wasatch region (Utah State in insert), with the Great Salt Lake (north) and Utah Lake (south) shown in blue and county borders in black. U.S. EPA designated non-attainment areas (NAA) for PM$_{2.5}$ are shown by red boundaries. From north to south these NAAs include the Logan NAA: "Moderate" status, Salt Lake City NAA: "Serious" status, and Provo NAA:**
**"Serious" status. UWFPS TO flight tracks are shown in pink. Purple markers indicate the locations of major cities, including Logan in Cache Valley, Ogden and Salt Lake City in SLV, and Provo in Utah Valley. The location of missed approaches conducted with the aircraft are shown by dark pink circles. The Hawthorne (HW) measurement site in SLV is labeled. (Right) Expanded view of SLV, with analyzed flight tracks highlighted in green.**



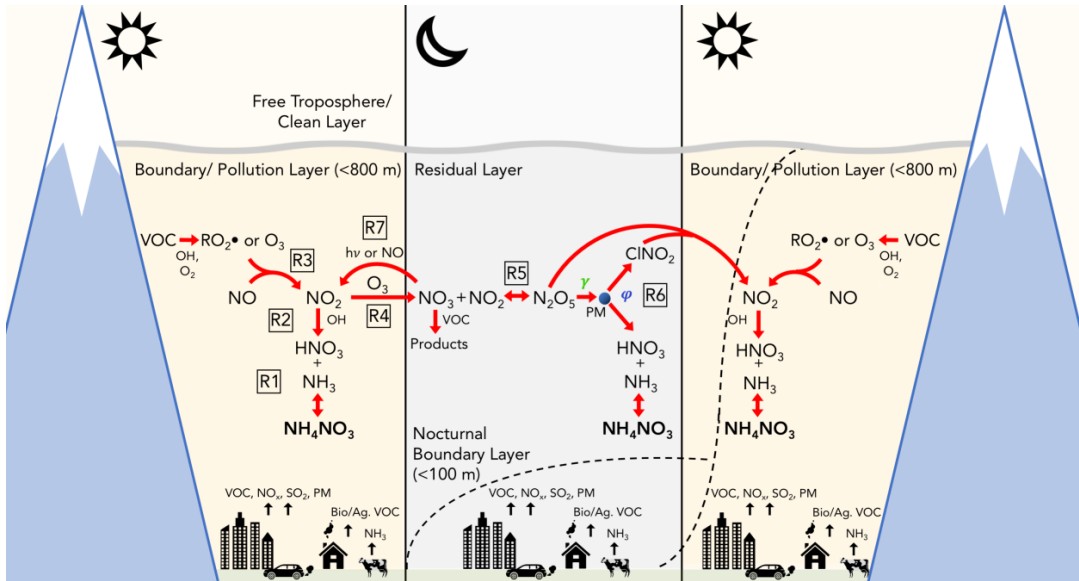

**Figure 2. Illustration of the day-night dynamics and chemical cycles of reactive nitrogen oxides, O₃, and NH₄NO₃ during PCAP conditions in SLV. The development of the nocturnal boundary layer and morning growth and mix-out are illustrated by the dashed lines. Figure is not to scale. (R6) represents the reaction:** $N_2O_5 \xrightarrow{\gamma(N_2O_5),\, M} 2*(1-\varphi)*HNO_3 + \varphi*ClNO_2$.

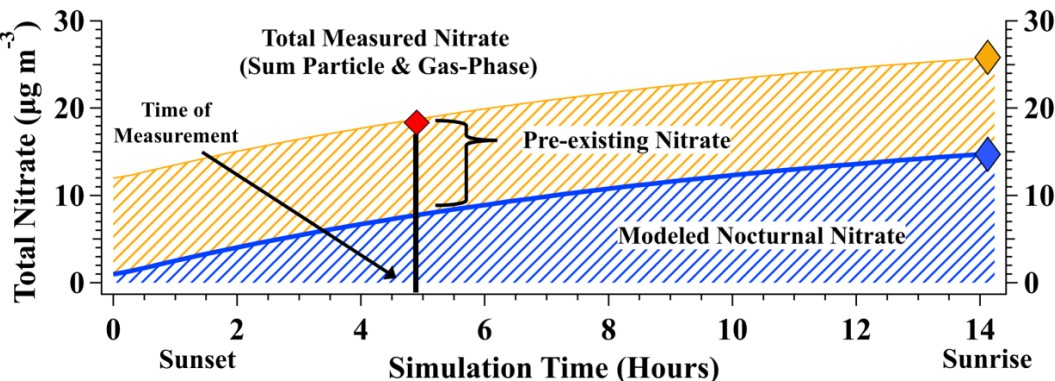

**Figure 3. Example simulation of total nitrate production from sunset to sunrise for an air parcel sampled over SLV on 28 January 2017.**
**Model derived $\gamma(N_2O_5)$ and $\phi(ClNO_2)$ values were 0.05 and 0.21, respectively. Modeled nocturnal nitrate (blue) is the total nitrate produced by heterogeneous chemistry in the box model, with the nocturnal production rate ($\mu g\ m^{-3}$ $night^{-1}$) represented by the blue diamond. Pre-existing nitrate (yellow) represents the nitrate present at sunset and is calculated as the difference between total measured nitrate from the aircraft (red diamond) and the model-predicted nitrate at the time of aircraft measurement (vertical black line). Assuming pre-existing nitrate is constant overnight (i.e. no deposition or dilution) and constant values of $\gamma(N_2O_5)$ and $\phi(ClNO_2)$, the**
**fractional contribution of nitrate production from a single night to the total observed is calculated as the ratio of modeled nitrate (blue diamond) to total nitrate (gold diamond) at sunrise.**





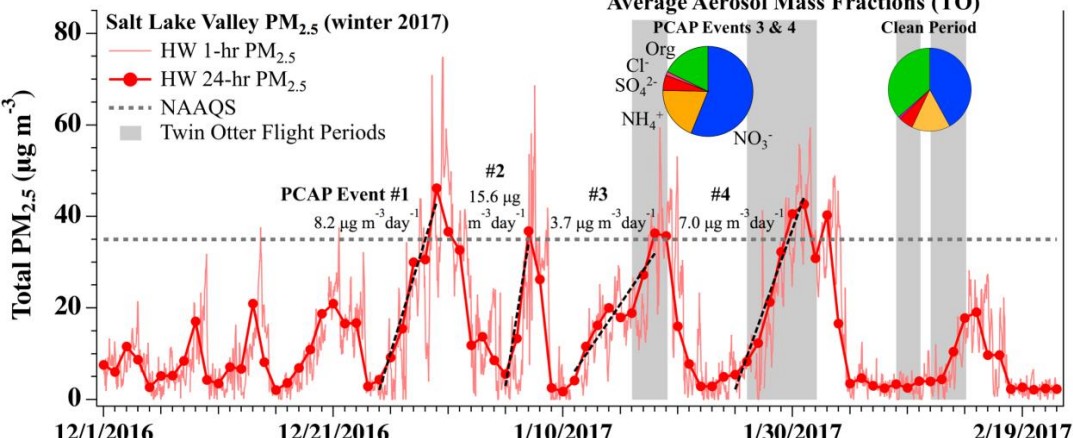

**Figure 4.** Time series of total PM$_{2.5}$ mass (µg m$^{-3}$) (1-hr and 24-hr averages) for the 2016-2017 winter, measured at the Hawthorne (HW) UDAQ site in SLV. Dashed black lines are daily PM$_{2.5}$ accumulation rates (rates given in Figure). The 24-hour EPA national ambient air quality standard for PM$_{2.5}$ (35 µg m$^{-3}$) is shown by the dashed gray line. Gray shading indicates days when the TO aircraft was flying during UWFPS. Average aerosol mass fractions measured by the AMS aboard the TO are given in pie charts for polluted and clean conditions. Aerosol components are colored by nitrate (blue), ammonium (gold), sulfate (red), non-refractory chloride (pink), and organics (green).

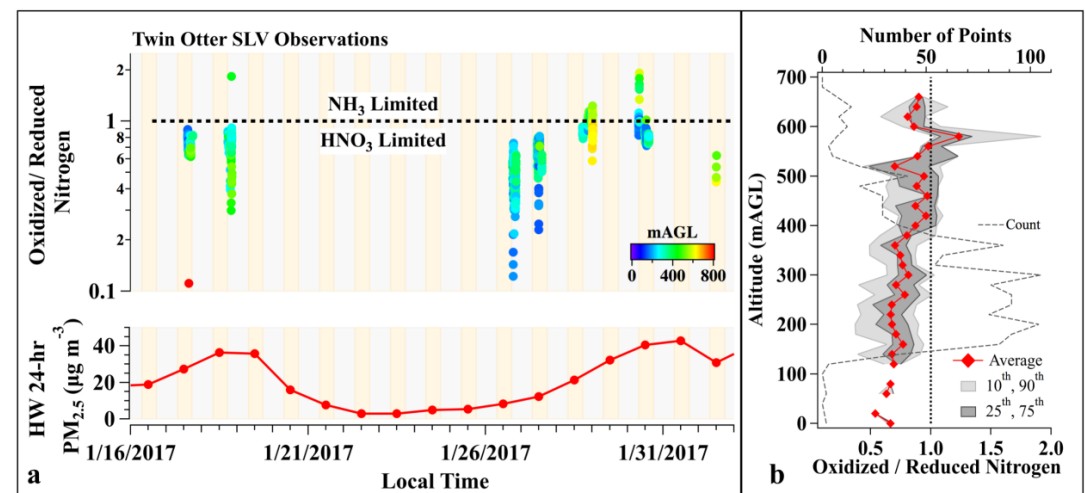

**Figure 5.** (a, top) Time series of ratio of total oxidized (HNO$_3$ + NO$_3^-$) to reduced (NH$_3$ + NH$_4^+$) nitrogen between 16 January and 1 February 2017 (10s averages), calculated from TO observations over SLV. Individual nitrogen ratios are colored by aircraft altitude (mAGL). Yellow and gray shading indicate times of day and night, respectively. (a, bottom) PM$_{2.5}$ mass (24-hour average) measured at the HW ground-site (bottom). (b) Vertical profile of oxidized to reduced nitrogen ratios from panel (a). Diamonds represent the average values in each altitude bin and gray shading shows the 10th-90th (light gray) and 25th-75th (dark gray) percentiles. The number of points in each bin is shown by the gray dashed line. The vertical black line illustrates a nitrogen ratio of 1.





**Figure 6. (a) Time series of NO₂, O₃ (top), $P_{NO_3^-}$ (middle, see text for definition), and PM₂.₅ (bottom) measured at the HW ground site during 16 January – 6 February 2017. O₃ data during the middle January pollution event were corrected to account for a 4.5 ppbv offset in the HW measurements, as shown in Figure S4. Aircraft flight times are shown by red shading. Dashed blue line shows the calculated**
5   **$P_{NO_3^-}$ rates that would occur during the day if this mechanism were operative. Solid blue line assumes nitrate production from this mechanism during the day is zero. Late afternoon $P_{NO_3^-}$ at the surface (dashed line), is roughly equivalent to the $P_{NO_3^-}$ expected in the RL at night. (b) Vertical profiles of O₃, NO₂, $P_{NO_3^-}$, and PM₁ measured from the aircraft on all night flights over SLV. In each panel, light shaded regions show the 10th-90th percentile ranges, dark shaded regions are the 25th-75th percentile ranges, and the solid lines are the 50th percentile. Dashed black lines show the number of points at each altitude.**



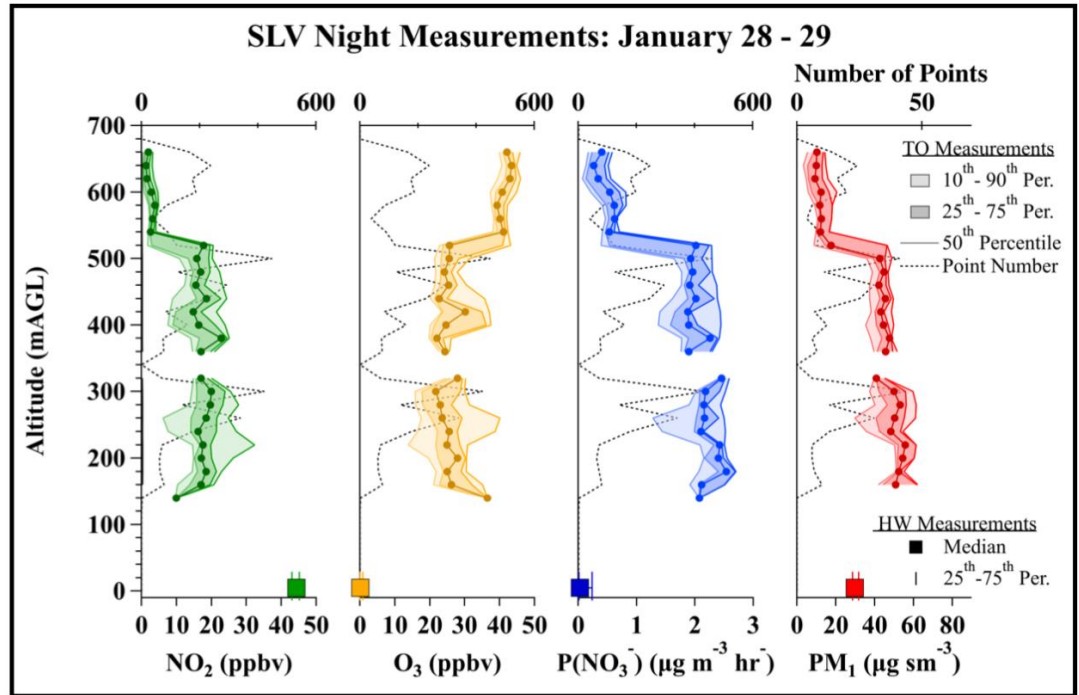

**Figure 7. Vertical Profiles of NO₂, O₃, P$_{NO_3^-}$, and PM₁ measured from the TO aircraft during 5 box patterns, flown over the SLV urban core between 21:20 and 00:30 MST on 28 and 29 January. Percentiles and number of points at each altitude are shown as in Figure 6. Square markers and error bars represent the median and 25th-75th percentile range of NO₂, O₃, P$_{NO_3^-}$, and PM₂.₅ measured concurrently at the HW ground site.**



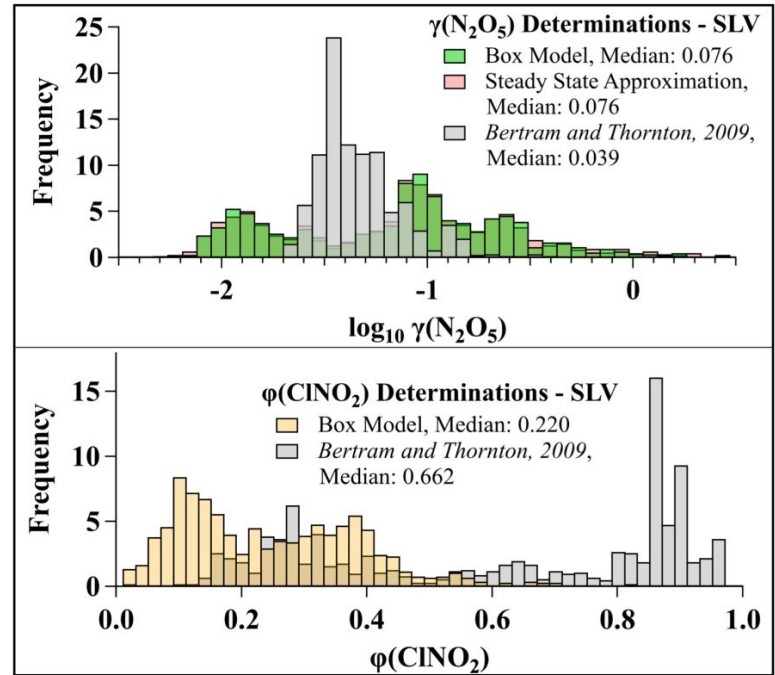

**Figure 8. (Top) Histograms of γ(N₂O₅) determinations from SLV during pollution events, calculated with the box model (green), steady state approximation (pink), and parameterization from Bertram and Thornton (2009). (Bottom) Histograms of φ(ClNO₂) determinations from SLV during pollution events calculated with the box model (gold) and parameterization from Bertram and Thornton (2009) (gray).**

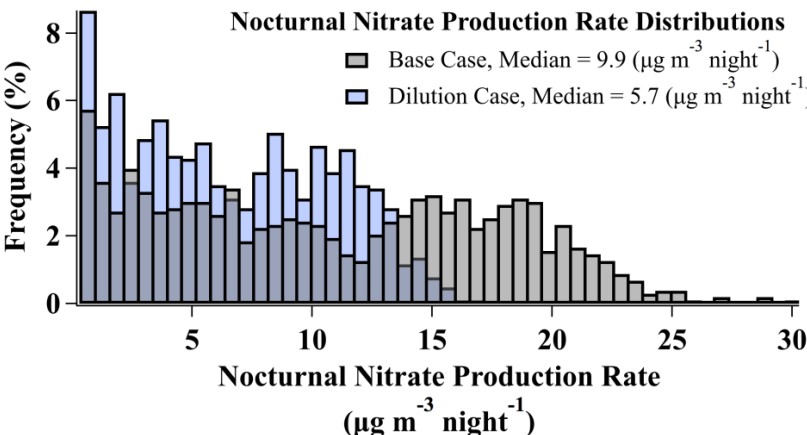

**Figure 9. Histograms of nocturnal nitrate production rates (µg m⁻³ night⁻¹) predicted by base case simulations and simulations incorporating a first-order dilution loss process with rate constant $k_{dilution}$ = 1.3×10⁻⁵ s⁻¹.**





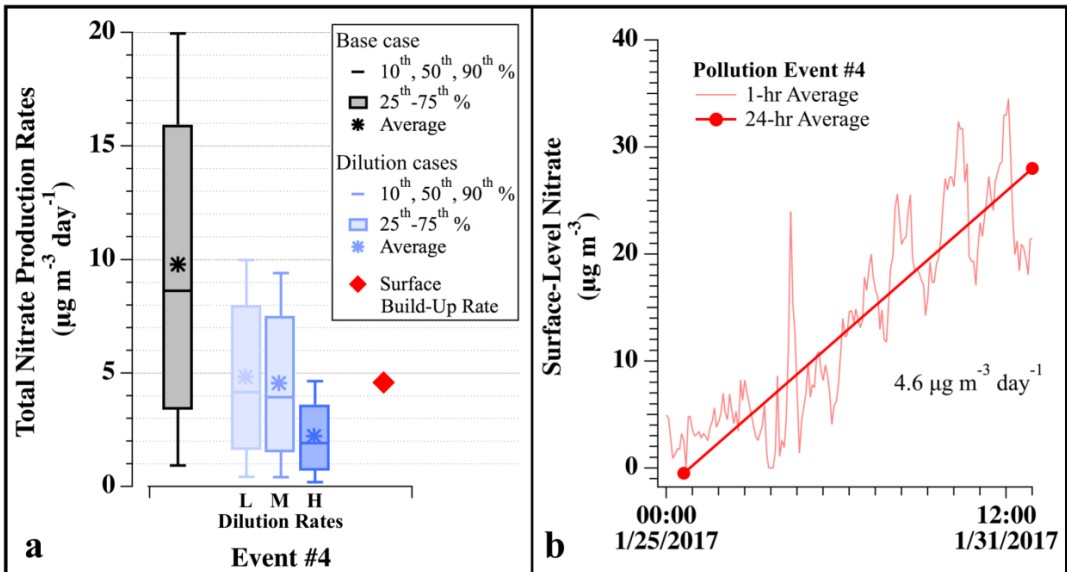

**Figure 10. (a) For pollution event #4, comparison of model-predicted nocturnal nitrate production ($\mu$g m$^{-3}$ day$^{-1}$) for base case simulations (gray), simulations with 24-hours of dilution (blue), and the average daily nitrate build-up observed at HW (red). Dilution cases are for simulations that incorporate nocturnal dilution rate constants of $1.2\times10^{-5}$ (L), $1.3\times10^{-5}$ (M), and $2.5\times10^{-5}$ (H) s$^{-1}$, scaled by 60% during the day. Box and whisker plots show the $10^{th} - 90^{th}$ percentile distributions of each set. The red diamond shows the ground-based build-up rate, calculated from 24-hr averaged data at HW in panel b. (b) Observed concentrations and average daily build-up rate of nitrate aerosol mass (total mass * 0.58) at HW during event #4.**



**Table 1. Aircraft measurements used in this analysis**

| Compound | Method / Instrument | Accuracy | Meas. Frequency | Location | Reference |
|---|---|---|---|---|---|
| ***Gas-Phase Species*** | | | | | |
| NO | CRDS[a] | 5% | 1s | Aircraft | (Fuchs et al., 2009; Wild et al., 2014) |
| $NO_2$ | CRDS | 5% | 1s | Aircraft | (Fuchs et al., 2009; Wild et al., 2014) |
| $O_3$ | CRDS | 5% | 1s | Aircraft | (Washenfelder et al., 2011; Wild et al., 2014) |
| $NO_y$ | CRDS | 12% | 1s | Aircraft | (Wild et al., 2014) |
| $N_2O_5$ | I⁻ToF-CIMS[c] | 30% | 1s | Aircraft | (Lee et al., 2014) |
| $ClNO_2$ | I⁻ToF-CIMS | 30% | 1s | Aircraft | (Lee et al., 2014) |
| $NH_3$ | QC-TILDAS[d] | | 1s | Aircraft | (Ellis et al., 2010) |
| ***Aerosol Measurements*** | | | | | |
| Aerosol (<1 μm) Composition | AMS[e] | 20% | 10s | Aircraft | (Bahreini et al., 2009; Middlebrook et al., 2012) |
| Dry Surface Area Density (<1 μm) | UHSAS[f] | 34%[g] | 3s | Aircraft | (Brock et al., 2011) |

[a]NOAA, Cavity Ring down Spectrometer (CRDS, NOxCaRD)
[b]Hawthorne
[c]University of Washington I⁻ Time of Flight Chemical Ionization Mass Spectrometer
[d]University of Toronto, Quantum Cascade Tunable Infrared Laser Differential Absorption Spectrometer
[e]NOAA, Aerosol Mass Spectrometer
[f]Droplet Measurement Techniques, Ultra-High Sensitivity Aerosol Spectrometer
[g]Estimated according to the performance of a different UHSAS in the WINTER campaign

