# Peer review of "On the contribution of nocturnal heterogeneous reactive nitrogen chemistry to particulate matter formation during wintertime pollution events in Northern Utah"

_Atmospheric Chemistry and Physics, 2019_

## Referee Comment (RC1) · Anonymous Referee #2 · 9 Apr 2019

Overall, I find this to be a very nice paper that addresses the important issue of particulate nitrate formation mechanisms in polluted valley regions. It is well written, with very nice figures, and a generally thorough analysis, although limited by non-consideration of daytime formation and most loss processes. I suggest that it be accepted once the authors address the issues below.

General Comments:

In the box model, does the loss rate of N2O5 have any impact on the O3 and NO2? If,

[Figure]

hypothetically, N2O5 loss is set to zero, then the O3 and NO2 would evolve differently than if the loss is fast, correct? The model, by separating the O3 and NO2 optimization and the N2O5 and ClNO2 optimization seems to neglect this. Is this a concern? Is the robustness tested by then using the derived gamma_N2O5 and ClNO2 yields to ensure that O3 and NO2 profiles are unchanged?

Fig. 6: For the night/day P(NO3) calculations, in panel a I find the legend descriptions to be a little confusing. Specifically, the meaning of "∼ RL at night" confuses that this is daytime.

Section 3.3.1: It seems like it would be a good idea to add a subscript (or some other indicator) of "max" to the P(NO3) values to make sure it is clear that these are the maximum. This is especially important for e.g. Fig. 6, where that context is not readily apparent.

Fig. 6b: Is there a reason that the number of points at a given altitude is so different between NO2, O3, and PNO3 (and PM1)? Does this have to do with estimation of the surface area and differences in averaging time?

P11/L33: Table S4 doesn't seem to address how uncertainties impact the ClNO2 yield, only net nitrate production. The paragraph, as written, sort of makes it seem like there is an implication of the ClNO2 yield being relatively insensitive to other uncertainties.

P12/Hygroscopic growth: Related, but not directly addressed, is that the time history of RH may matter. The authors use the RH at the point of measurement and, somehow, extrapolate this back in time in the model to give a minute-by-minute perspective on particle surface area. Historical fluctuations in RH could influence the point observations. Is there an attempt to account for variations in RH with time of day? Similarly, the authors seem to use a fixed dry SA, based on the intercept point. But if aerosol growth is occurring then the SA would evolve over time. What sort of uncertainty does this simplification bring in?

P12/L13: It should also be noted that the steady state approximation requires the surface area and temperature, not just NO2, O3, and N2O5.

P13/L33: It would be good if the authors commented here on the substantial variability in the derived values and, perhaps, the seeming bimodality (with a high production mode and a lower production mode).

P14: The discussion of dilution/entrainment could be enhanced. The Womak paper was just published on 4/8, making it available. It seems that the authors here are arguing for an entrainment rate that is largely independent of time of day, except for the issue of changing height of the mixed layer. However, one might expect the entrainment rate to differ notably between the daytime and nighttime. It is unclear whether the authors are applying a daytime entrainment rate to the nighttime or really, in general, how entrainment is being accounted for. The origin of the 40% scaling factor is somewhat mysterious as well.

P15/L3: The difference between the L and M cases seems negligible, as is evident from Fig. 10. Why include both of these when they are so similar?

P15/L4: Should the transition from nighttime to daytime also be accounted for? In other words, if most nitrate is formed aloft (as suggested) and the air in the RL is entrained to the surface starting at sunrise, then the nitrate in the RL will be distributed throughout the daytime mixed layer. Dilution from exchange with the FT will occur on top of this. However, that would also require accounting for the nitrate in the surface layer initially.

P15/L9: I have some difficulty with the framing here. The authors start by saying that nocturnal chemistry and largely explain the nitrate accumulation at the surface. But then they go on to say that Womack (who looked at the same events) concluded that photochemical production is quite important too. Notably, while Pusede et al. (2016) implicated nocturnal nitrate production as very important, they did not discount daytime production to the extent suggested here and Prabhakar et al. (2017), building on Pusede et al. (2016), concluded that daytime production plays an important role. I

suggest that the authors consider revising the first sentence and how they frame the discussion here. They have only considered nighttime formation and thus their ultimate conclusions regarding the contributions of daytime processes are limited.

Minor points:

A minor grammar question: should it be "in the SLV" or "in SLV". I would have thought the former, as the SLV is not a city but a region (like the western US, which the authors refer to with a "the").

P13/L5: "empirically-based" should just be "empirical."
* * *

---

## Referee Comment (RC2) · Anonymous Referee #1 · 6 May 2019

McDuffie et al., presented an observed constrain analysis about the particulate nitrate production over vertical scale in the Northern Utah Valley. They found large amount of nitrate was produced aloft due to the air mass is free from the titration effect by the emitted NO near the surface. Although N2O5 uptake coefficient in this study is much higher than previous winter studies in US, the nocturnal particulate nitrate production rate is not limited by heterogeneous hydrolysis but the oxidation of NO2 by O3. Take the consideration of the nocturnal dilution and daytime entrainment, the model predicted nocturnal nitrate production in residual layer dominates the increasing of nitrate

in the diagnosed polluted episode, and highlights future work should considering these processes. This study is very important to the community for recognizing the winter particulate nitrate pollution by heterogeneous reaction not only in surface layer but also above the canopy of the urban/suburban (similar results also obtained in Beijing based on tower measurements https://doi.org/10.5194/acp-18-10483-2018 which is worth to be cited in this paper). This paper certainly worth to be published on ACP subject to minor revision.

1. Section 3.3.1, I can understand what the authors want to present here, but I strongly suggest changing the PN2O5 to PNO3. for the convenience of readers who not so familiar with NO3 chemistry, otherwise it is hard to get the point of Eq. 6.

2. The derived N2O5 uptake coefficient is high that previous two studies conducted by the same group though the iterative box model, if the N2O5 uptake efficiency is high enough and the production rate of particulate nitrate is only limited by the NO2 + O3, N2O5 concentration should be low, could the author provide more information about observed N2O5 concentration?

3. The label in Figure S2(b) is inconsistent with the description in the main text, where the median dry SA should be 151.9 ug m-3.

4. Page 8, line 7, missed a subscript the (NH4)2SO4

5. The production rate of particulate nitrate in Figure 6 and Figure 7 should be united in the main text as PNO3-. Figure 6b the unit of P(NO3-) and PM1.0 should be corrected.

6. SI, Section S2 PNO3- Calculation Details, repeated "in" (In in Section 3.3.1 of this analysis)

---

## Author Comment (AC1) · 15 Jun 2019

**Response to Referees**

We thank the referees for their comments. They have helped improve the clarity and quality of the manuscript. We have addressed each comment as follows: the original comments are shown in black, our responses are in blue, and corresponding changes to the manuscript are in *blue italics*. Any additional changes to the manuscript correct typographical errors and do not change the content.

**Response to Referee #1**

This study is very important to the community for recognizing the winter particulate nitrate pollution by heterogeneous reaction not only in surface layer but also above the canopy of the urban/suburban (similar results also obtained in Beijing based on tower measurements https://doi.org/10.5194/acp-18-10483-2018 which is worth to be cited in this paper).

The cited paper highlights both the importance of  $N_2O_5$  particle nitrate formation aloft, as well as the insensitivity of particle nitrate formation in polluted regions to changes in  $\gamma(N_2O_5)$  at sufficiently large values. Citations have been added to this paper accordingly.

**Page 3, Line 10**

Similarly, a box model analysis of tower and ground-based observations in Beijing, China also identified these processes as important contributors to surface-level particulate nitrate the following day (Wang et al., 2018).

**Page 13, Line 22**

Despite disagreement between the box model and parameterizations, the  $\gamma(N_2O_5)$  values predicted by all three methods are large enough, in combination with the large measured aerosol SA, to fall within the range where models of nighttime chemistry are insensitive to variation in uptake efficiency (e.g. Macintyre and Evans, 2010; Riemer et al., 2003; Wang et al., 2018).

Section 3.3.1, I can understand what the authors want to present here, but I strongly suggest changing the PN2O5 to PNO3. for the convenience of readers who not so familiar with NO3 chemistry, otherwise it is hard to get the point of Eq. 6.

In order to maintain consistent terminology with previous studies (e.g. Bassandorj et al., 2017) and to help clarify this section, we have changed  $P_{N_2O_5}$  to  $P_{NO_3}$  as the referee suggests. Please see our response to referee #2 for our full response related to this change.

The derived N2O5 uptake coefficient is high that previous two studies conducted by the same group though the iterative box model, if the N2O5 uptake efficiency is high enough and the production rate of particulate nitrate is only limited by the NO2 + O3, N2O5 concentration should be low, could the author provide more information about observed N2O5 concentration?

We have added a supplemental figure S6 showing the vertical distribution of N2O5, ClNO2,  $\gamma$ (N2O5), and  $\phi$ (ClNO2) during night flights over the SLV. We have also edited the following sentence in the main text to provide additional information about the N2O5 observations:

**Page 13, Line 26**

As further evidence of this limitation, the median lifetime of NO2 with respect to O3 ( $\tau_{NO_2} = 1/(k_1[O_3])$ ) was 9-hours while the lifetime of N2O5 ( $\tau_{N_2O_5} = 1/k_{N_2O_5}$ ) was just 14-minutes, resulting in low N2O5 mixing ratios (median = 0.03 ppbv) during the SLV pollution events (Figure S6).

Supplemental Figure S6.

Figure S6. Vertical profiles of  $N_2O_5$ , CINO2 (1-second measurements), and box-model derived  $\gamma(N_2O_5)$  and  $\phi(CINO_2)$  values from all night flights over the SLV. In each panel, light shaded regions show the  $10^{th}$ - $90^{th}$  percentile ranges, dark shaded regions are the  $25^{th}$ - $75^{th}$  percentile ranges, and the solid lines are the  $50^{th}$  percentile. Dashed black lines show the number of points at each altitude.

The label in Figure S2(b) is inconsistent with the description in the main text, where the median dry SA should be 151.9 ug m-3.

We have changed 151 to 151.9 and 353 to 353.1 to be consistent with Figure S2b.

Page 6, Line 7 For the 1031 10-second measurement periods with simultaneous values of  $\gamma(N_2O_5)$  and  $\phi(CINO_2)$ , the median dry aerosol SA was 151.9  $\mu m^2$  cm-3, which increased to 353.1  $\mu m^2$  cm-3 when accounting for hygroscopic growth (Figure S2b).

Page 8, line 7, missed a subscript the (NH4)2SO4

**Corrected**

The production rate of particulate nitrate in Figure 6 and Figure 7 should be united in the main text as PNO3-. Figure 6b the unit of P(NO3-) and PM1.0 should be corrected.

We have changed the  $P(NO_3^{-1})$  label in Figures 6 and 7 (see figure updates in our response to referee #2) to  $P_{NO_3^{-1},max}$  to reflect and maintain consistency with the changes made in the text in response to referee #2. We also fixed a typo in its unit label. We have also updated  $PM_1$  data in Figures 6 and 7 from units of ug sm-3 to display units of ug m-3 in order to maintain consistency with other parameters discussed here. This change impacts Figures 6b and 7 only, not the discussion or analyses.

SI, Section S2 PNO3- Calculation Details, repeated "in" (In in Section 3.3.1 of this analysis)

**Corrected**

**Response to Referee #2**

In the box model, does the loss rate of N2O5 have any impact on the O3 and NO2? If, hypothetically, N2O5 loss is set to zero, then the O3 and NO2 would evolve differently than if the loss is fast, correct? The model, by separating the O3 and NO2 optimization and the N2O5 and CINO2 optimization seems to neglect this. Is this a concern? Is the robustness tested by then using the derived gamma\_N2O5 and CINO2 yields to ensure that O3 and NO2 profiles are unchanged?

The referee is correct that the loss rate of N2O5 will impact the evolution of O3 and NO2. Due to this dependence, the model does not completely separate the derivation of initial NO2 and O3 from the derivation of  $k_{N_2O_5}$ . Rather, once  $k_{N_2O_5}$  has been derived, the model re-iterates both processes, re-calculating the initial O3 and NO2 concentrations with the updated  $k_{N_2O_5}$  value, and then re-calculating  $k_{N_2O_5}$ . This entire process repeats until model predicted concentrations of O3, NO2, and N2O5 simultaneously reproduce the observed values. As  $k_{CINO_2}$  does not impact the evolution of O3, NO2, or the total loss rate of N2O5 ( $k_{N_2O_5}$ ), the third step derives  $k_{CINO_2}$  by iteratively fitting to CINO2 observations. We have clarified the extent of the model iteration in the following text.

**Page 5, Line 5**

Briefly, the model forward-integrates the chemical mechanism (13 reactions, Table S1) starting 1.3 hours prior to sunset (see below), iteratively adjusting the initial concentrations of  $O_3$  and  $NO_2$ , until the model-predicted concentrations are both within 0.5% of the aircraft observations. Holding these initial concentrations constant, the model next adjusts the total heterogeneous loss rate constant of  $N_2O_5$  ( $k_{N_2O_5}$ ) until the model output reproduces ambient nighttime observations of  $N_2O_5$  to within 1%. As described in McDuffie et al. (2018b), the model iterates these steps, re-adjusting initial concentrations of  $O_3$  and  $NO_2$  and values of  $k_{N_2O_5}$  until aircraft observations of  $NO_2$ ,  $O_3$ , and  $N_2O_5$  are simultaneously reproduced by the model. The final step holds these values constant while iteratively adjusting the production rate of  $CINO_2$  ( $k_{CINO_2}$ ) until the modeled mixing ratios of  $CINO_2$  are within 1% of the nighttime  $CINO_2$  observations.

Fig. 6: For the night/day P(NO3) calculations, in panel a I find the legend descriptions to be a little confusing. Specifically, the meaning of " $\sim$  RL at night" confuses that this is daytime.

We have changed the label to:

" $P_{(NO_3^-,max)}$ : day only ~ nighttime RL" and have kept the original description in the Figure caption.

Fig. 6b: Is there a reason that the number of points at a given altitude is so different between NO2, O3, and PNO3 (and PM1)? Does this have to do with estimation of the surface area and differences in averaging time?

We appreciate the referee's attention to detail. In the original figure we had incorrectly plotted the NO2 and O3 data from all flights in the SLV, not just the data collected at night. We have updated the vertical profiles of these species in Figure 6b as shown below. As  $P_{(NO_3^-,max)}$  is calculated using NO2, O3, temperature, and pressure, the number of  $P_{(NO_3^-,max)}$  determinations is now equivalent to the number of NO2 and O3 measurements. The number of PM1 measurements is lower due to the difference in measurement frequency of the AMS. We have now indicated this difference in the figure captions of Figure 6 and 7 (below). This error did not carry over to the analysis and does not impact the discussions in the main text or supplement. The vertical profiles in Figure 7 have also been checked and are correct. In addition, the PM1 data in Figure 6b (and Figure 7) have been changed from units of ug sm-3 to ug m-3 in response to suggestions by referee #1.

Relevant changes to Figure 6 caption:

(b) Vertical profiles of  $O_3$ ,  $NO_2$ ,  $P_{NO_3^-,max}$  (1-second data) and  $PM_1$  (10-second data) measured from the aircraft on all night flights over the SLV.

**Relevant changes to Figure 7 caption:**

*Vertical Profiles of NO*2, *O*3,  $P_{NO_3^-,max}$  (1-second data), and PM1 (10-second data) measured from the TO aircraft during 5 box patterns, flown over the SLV urban core between 21:20 and 00:30 MST on 28 and 29 January.

---

## Author Response (AR1)

**Response to Referees**

We thank the referees for their comments. They have helped improve the clarity and quality of the manuscript. We have addressed each comment as follows: the original comments are shown in black, our responses are in blue, and corresponding changes to the manuscript are in *blue italics*. Any additional changes to the manuscript correct typographical errors and do not change the content.

**Response to Referee #1**

This study is very important to the community for recognizing the winter particulate nitrate pollution by heterogeneous reaction not only in surface layer but also above the canopy of the urban/suburban (similar results also obtained in Beijing based on tower measurements https://doi.org/10.5194/acp-18-10483-2018 which is worth to be cited in this paper).

The cited paper highlights both the importance of $N_2O_5$ particle nitrate formation aloft, as well as the insensitivity of particle nitrate formation in polluted regions to changes in $\gamma(N_2O_5)$ at sufficiently large values. Citations have been added to this paper accordingly.

*Page 3, Line 10*
*Similarly, a box model analysis of tower and ground-based observations in Beijing, China also identified these processes as important contributors to surface-level particulate nitrate the following day (Wang et al., 2018).*

*Page 13, Line 22*
*Despite disagreement between the box model and parameterizations, the $\gamma(N_2O_5)$ values predicted by all three methods are large enough, in combination with the large measured aerosol SA, to fall within the range where models of nighttime chemistry are insensitive to variation in uptake efficiency (e.g. Macintyre and Evans, 2010; Riemer et al., 2003; Wang et al., 2018).*

Section 3.3.1, I can understand what the authors want to present here, but I strongly suggest changing the PN2O5 to PNO3. for the convenience of readers who not so familiar with NO3 chemistry, otherwise it is hard to get the point of Eq. 6.

In order to maintain consistent terminology with previous studies (e.g. Bassandorj et al., 2017) and to help clarify this section, we have changed $P_{N_2O_5}$ to $P_{NO_3}$ as the referee suggests. Please see our response to referee #2 for our full response related to this change.

The derived N2O5 uptake coefficient is high that previous two studies conducted by the same group though the iterative box model, if the N2O5 uptake efficiency is high enough and the production rate of particulate nitrate is only limited by the NO2 + O3, N2O5 concentration should be low, could the author provide more information about observed N2O5 concentration?

We have added a supplemental figure S6 showing the vertical distribution of $N_2O_5$, $ClNO_2$, $\gamma(N_2O_5)$, and $\phi(ClNO_2)$ during night flights over the SLV. We have also edited the following sentence in the main text to provide additional information about the $N_2O_5$ observations:

*Page 13, Line 26*
*As further evidence of this limitation, the median lifetime of NO$_2$ with respect to O$_3$ ($\tau_{NO_2} = 1/(k_1[O_3])$) was 9-hours while the lifetime of N$_2$O$_5$ ($\tau_{N_2O_5} = 1/k_{N_2O_5}$) was just 14-minutes, resulting in low N$_2$O$_5$ mixing ratios (median = 0.03 ppbv) during the SLV pollution events (Figure S6).*

*Supplemental Figure S6.*

[Figure]

**Figure S6. Vertical profiles of $N_2O_5$, $ClNO_2$ (1-second measurements), and box-model derived $\gamma(N_2O_5)$ and $\phi(ClNO_2)$ values from all night flights over the SLV. In each panel, light shaded regions show the $10^{th}$-$90^{th}$ percentile ranges, dark shaded regions are the $25^{th}$-$75^{th}$ percentile ranges, and the solid lines are the $50^{th}$ percentile. Dashed black lines show the number of points at each altitude.**

The label in Figure S2(b) is inconsistent with the description in the main text, where the median dry SA should be 151.9 ug m-3.

We have changed 151 to 151.9 and 353 to 353.1 to be consistent with Figure S2b.

*Page 6, Line 7*
*For the 1031 10-second measurement periods with simultaneous values of $\gamma(N_2O_5)$ and $\phi(ClNO_2)$, the median dry aerosol SA was 151.9 $\mu m^2\ cm^{-3}$, which increased to 353.1 $\mu m^2\ cm^{-3}$ when accounting for hygroscopic growth (Figure S2b).*

Page 8, line 7, missed a subscript the (NH4)2SO4

Corrected

The production rate of particulate nitrate in Figure 6 and Figure 7 should be united in the main text as PNO3-. Figure 6b the unit of P(NO3-) and PM1.0 should be corrected.

We have changed the P($NO_3^-$) label in Figures 6 and 7 (see figure updates in our response to referee #2) to $P_{NO_3^-,max}$ to reflect and maintain consistency with the changes made in the text in response to referee #2. We also fixed a typo in its unit label. We have also updated $PM_1$ data in Figures 6 and 7 from units of ug sm$^{-3}$ to display units of ug m$^{-3}$ in order to maintain consistency with other parameters discussed here. This change impacts Figures 6b and 7 only, not the discussion or analyses.

SI, Section S2 PNO3- Calculation Details, repeated "in" (In in Section 3.3.1 of this analysis)

Corrected

**Response to Referee #2**

In the box model, does the loss rate of N2O5 have any impact on the O3 and NO2? If, hypothetically, N2O5 loss is set to zero, then the O3 and NO2 would evolve differently than if the loss is fast, correct? The model, by separating the O3 and NO2 optimization and the N2O5 and ClNO2 optimization seems to neglect this. Is this a concern? Is the robustness tested by then using the derived gamma_N2O5 and ClNO2 yields to ensure that O3 and NO2 profiles are unchanged?

The referee is correct that the loss rate of $N_2O_5$ will impact the evolution of $O_3$ and $NO_2$. Due to this dependence, the model does not completely separate the derivation of initial $NO_2$ and $O_3$ from the derivation of $k_{N_2O_5}$. Rather, once $k_{N_2O_5}$ has been derived, the model re-iterates both processes, re-calculating the initial $O_3$ and $NO_2$ concentrations with the updated $k_{N_2O_5}$ value, and then re-calculating $k_{N_2O_5}$. This entire process repeats until model predicted concentrations of $O_3$, $NO_2$, and $N_2O_5$ simultaneously reproduce the observed values. As $k_{ClNO_2}$ does not impact the evolution of $O_3$, $NO_2$, or the total loss rate of $N_2O_5$ ($k_{N_2O_5}$), the third step derives $k_{ClNO_2}$ by iteratively fitting to $ClNO_2$ observations. We have clarified the extent of the model iteration in the following text.

*Page 5, Line 5*
*Briefly, the model forward-integrates the chemical mechanism (13 reactions, Table S1) starting 1.3 hours prior to sunset (see below), iteratively adjusting the initial concentrations of O₃ and NO₂, until the model-predicted concentrations are both within 0.5% of the aircraft observations. Holding these initial concentrations constant, the model next adjusts the total heterogeneous loss rate constant of N₂O₅ (kₙ₂ₒ₅) until the model output reproduces ambient nighttime observations of N₂O₅ to within 1%. As described in McDuffie et al. (2018b), the model iterates these steps, re-adjusting initial concentrations of O₃ and NO₂ and values of kₙ₂ₒ₅ until aircraft observations of NO₂, O₃, and N₂O₅ are simultaneously reproduced by the model. The final step holds these values constant while iteratively adjusting the production rate of ClNO₂ (k_{ClNO₂}) until the modeled mixing ratios of ClNO₂ are within 1% of the nighttime ClNO₂ observations.*

Fig. 6: For the night/day P(NO3) calculations, in panel a I find the legend descriptions to be a little confusing. Specifically, the meaning of "~ RL at night" confuses that this is daytime.

We have changed the label to:
"$P_{(NO_3^-,max)}$: *day only ~ nighttime RL*"
and have kept the original description in the Figure caption.

Fig. 6b: Is there a reason that the number of points at a given altitude is so different between NO2, O3, and PNO3 (and PM1)? Does this have to do with estimation of the surface area and differences in averaging time?

We appreciate the referee's attention to detail. In the original figure we had incorrectly plotted the $NO_2$ and $O_3$ data from all flights in the SLV, not just the data collected at night. We have updated the vertical profiles of these species in Figure 6b as shown below. As $P_{(NO_3^-,max)}$ is calculated using $NO_2$, $O_3$, temperature, and pressure, the number of $P_{(NO_3^-,max)}$ determinations is now equivalent to the number of $NO_2$ and $O_3$ measurements. The number of $PM_1$ measurements is lower due to the difference in measurement frequency of the AMS. We have now indicated this difference in the figure captions of Figure 6 and 7 (below). This error did not carry over to the analysis and does not impact the discussions in the main text or supplement. The vertical profiles in Figure 7 have also been checked and are correct. In addition, the $PM_1$ data in Figure 6b (and Figure 7) have been changed from units of ug sm⁻³ to ug m⁻³ in response to suggestions by referee #1.

Relevant changes to Figure 6 caption:
*(b) Vertical profiles of O₃, NO₂, P_{NO₃⁻,max} (1-second data) and PM₁ (10-second data) measured from the aircraft on all night flights over the SLV.*

Relevant changes to Figure 7 caption:
*Vertical Profiles of $NO_2$, $O_3$, $P_{NO_3^-,max}$ (1-second data), and $PM_1$ (10-second data) measured from the TO aircraft during 5 box patterns, flown over the SLV urban core between 21:20 and 00:30 MST on 28 and 29 January.*

Figure 6
Original Version
[Figure]
 Updated Version

[Figure]

 Section 3.3.1: It seems like it would be a good idea to add a subscript (or some other indicator) of "max" to the P(NO3) values to make sure it is clear that these are the maximum. This is especially important for e.g. Fig. 6, where that context is not readily apparent.

We have changed $P_{NO_3^-}$ to $P_{NO_3^-,max}$ to help distinguish $P_{NO_3}$ and $P_{NO_3^-}$, as well as remind the reader that $P_{NO_3^-,max}$ values are upper limit estimates for the production rate of particulate nitrate from $N_2O_5$ heterogeneous chemistry. We have also changed $P_{N_2O_5}$ to $P_{NO_3}$ following the suggestion of referee #1. We have slightly changed the first paragraph is Section 3.3.1 as follows and changed the $P_{N_2O_5}$ and $P_{NO_3^-}$ terminology throughout the main text, supplement, and Figures 6 and 7.

*Page 9, Line 8*
*An upper limit estimate of the instantaneous production rate of aerosol nitrate from heterogeneous $N_2O_5$ chemistry is defined here as $P_{NO_3^-,max}$. This rate can be calculated as two times the gas-phase production rate of the $NO_3$ radical ($P_{NO_3}$), given that reaction between $NO_2$ and $O_3$ (Eqs. (4) – (6)), rather than $N_2O_5$ uptake,  is the rate limiting step for nitrate formation (discussed below). In Eq. (4), $P_{NO_3}$ is calculated in units of molec. $cm^{-3}$ $s^{-1}$ but is typically reported in units of ppbv $hr^{-1}$ as shown below. The reaction kinetics in Eq. (5) between $NO_2$ and $O_3$ are from the 2008 IUPAC recommendation (IUPAC, 2008) and  $P_{NO_3^-,max}$  in Eq. (6) is calculated after $P_{NO_3}$ has been converted to units of $\mu g$ $m^{-3}$ $hr^{-1}$, as detailed in Supplemental Section S2. This calculation estimates a maximum contribution of $N_2O_5$ heterogeneous chemistry to nitrate production as it assumes: 1) $N_2O_5$ is produced quantitatively from $NO_3$ (i.e. no competing reaction of $NO_3 + VOC$), 2) $N_2O_5$ is produced at the rate of $NO_3$ production (valid under cold conditions that shift the $NO_3$-$N_2O_5$ equilibrium to favor of $N_2O_5$), 3) $N_2O_5$ is efficiently taken up onto aerosol, and 4) aqueous-phase reactions form two molecules of $HNO_3$ for every molecule of $N_2O_5$ (i.e. $\phi(ClNO_2) = 0$).*

$$P_{NO_3}[ppbv\ hr^{-1}] = \frac{k_4[O_3][NO_2]}{ND\ [molec.\ cm^{-3}]} * 3600\ [s\ hr^{-1}] * 1 \times 10^9\ [ppbv] \qquad (1)$$

$$k_4\ [cm^3\ molecule^{-1}\ s^{-1}] = 1.4 \times 10^{-13} e^{(-2470/T)} \qquad (2)$$

$$P_{NO_3^-,max}[\mu g\ m^{-3}\ hr^{-1}] = 2 * (P_{NO_3}\ [\mu g\ m^{-3}\ hr^{-1}]) \qquad (3)$$

Figure 6 changes – see above

Figure 7 changes
Original Version                                     Updated Version

[Figure]

P11/L33: Table S4 doesn't seem to address how uncertainties impact the ClNO2 yield, only net nitrate production. The paragraph, as written, sort of makes it seem like there is an implication of the ClNO2 yield being relatively insensitive to other uncertainties.

This paragraph discusses uncertainties in both $k_{N_2O_5}$ and $k_{ClNO_2}$ that may lead to their respective over- and under-predictions. In Table S4, we have chosen to only show sensitivities of nitrate production (rather than $\gamma(N_2O_5)$ and $\phi(ClNO_2)$ individually) because 1) uncertainties in nitrate production will be impacted by uncertainties in both $k_{N_2O_5}$ and $k_{ClNO_2}$, 2) nitrate production is the main focus of this study, and 3) our analysis shows that $\gamma(N_2O_5)$ values are large enough that nitrate production will be largely be insensitive to changes in $\gamma(N_2O_5)$. To highlight that the nitrate production sensitivities also include uncertainties in $k_{ClNO_2}$, we have re-phrased the last sentence in this paragraph as follows:

*Page 11, Line 37*
*Overall, while the box model has a large number of uncertainties and assumptions, predictions of nocturnal nitrate production, which are subject to uncertainties in both $k_{N_2O_5}$ and $k_{ClNO_2}$, are not highly sensitive to sources other than dilution (discussed below, Table S4).*

P12/Hygroscopic growth: Related, but not directly addressed, is that the time history of RH may matter. The authors use the RH at the point of measurement and, somehow, extrapolate this back in time in the model to give a minute-by-minute perspective on particle surface area. Historical fluctuations in RH could influence the point observations. Is there an attempt to account for variations in RH with time of day? Similarly, the authors seem to use a fixed dry SA, based on the intercept point. But if aerosol growth is occurring then the SA would evolve over time. What sort of uncertainty does this simplification bring in?

There are two separate issues relating to hygroscopic growth that we would like to clarify.

First is the possibility that we discuss on page 12 where an under-estimation of wet aerosol SA *at the time of measurement* is possible due to uncertainties in the hygroscopic growth curve at high RH. This would not impact the model derivation of $k_{N_2O_5}$ but would cause the $\gamma(N_2O_5)$ value from Eq. 1 to be too high.

The second issue that the referee raises is about the time-varying evolution of relative humidity and aerosol surface area between sunset and the time of measurement. Our box model assumes that sampled air parcels evolve over-night with constant temperature (i.e. constant reaction rate constants) and relative humidity (i.e. constant hygroscopic growth factor/wet SA). Therefore, the model-derived values ($k_{N_2O_5}$ and $k_{ClNO_2}$, $\gamma(N_2O_5)$ and $\phi(ClNO_2)$ by extension) are representative of the average conditions that lead to the observed concentrations of the model fit parameters, $NO_2$, $O_3$, $N_2O_5$, and $ClNO_2$. We agree that the history of RH and aerosol mass accumulation is important to consider and may result in a time-dependence of $k_{N_2O_5}$, that has been discussed previously in McDuffie et al., 2018b. As this time-dependence will not impact the potential under-estimation of calculated wet SA discussed on page 12, we have not made any changes to this section of the manuscript. Instead, we have added a brief discussion in the previous paragraph about the assumptions of constant T and SA. In addition, we have estimated the potential growth in aerosol surface area overnight using the model estimates of aerosol nitrate mass production. We have made no further changes as we have found that absolute nitrate production, the focus of this study, is not limited by $k_{N_2O_5}$ but by gas-phase oxidation rates.

The following changes have been made:

*Page 5, Line 3*
*A zero-dimension chemical box model has been developed to simulate the nocturnal chemical evolution of an air parcel from sunset until the time of aircraft measurement (assuming constant temperature and relative humidity).*

*Page 6, Line 9*
*Additional uncertainties associated with hygroscopic growth and assumptions of constant SA are discussed below in Section 3.3.2.*

*Page 11, Line 33*
*Additional uncertainties in $k_{N_2O_5}$ and $k_{ClNO_2}$ may arise from model assumptions of constant temperature and RH (i.e. rate constants and surface area) overnight. While model sensitivities to these uncertainties cannot be directly quantified, the percent growth in SA from nitrate accumulation is estimated to be less than the uncertainty in the dry SA measurement (34%). As modeled $k_{N_2O_5}$ values are also consistent with those derived from observations (discussed below), this source of uncertainty is not discussed further.*

P12/L13: It should also be noted that the steady state approximation requires the surface area and temperature, not just NO2, O3, and N2O5.

Changed

*Page 12, Line 19*
*The first method calculates $\gamma(N_2O_5)$ from observations of temperature, SA, $NO_2$, $O_3$, and $N_2O_5$, based on the steady state approximation ($\gamma(N_2O_5)_{ss}$), described by Brown et al. (2003) and defined in Supplemental Section S4.1.*

P13/L33: It would be good if the authors commented here on the substantial variability in the derived values and, perhaps, the seeming bimodality (with a high production mode and a lower production mode).

We do not see a bimodality in Figure 9 (the nocturnal production rate of nitrate) and rather assume that the referee is referring to the two apparent modes in the $N_2O_5$ uptake coefficients in Figure 8. The source of these two $\gamma(N_2O_5)$ modes has not been investigated. Our analysis has shown that the production of aerosol nitrate (the focus of this study) is limited by the gas-phase oxidation of $NO_2$ ($P_{NO_3}$) rather than $N_2O_5$ uptake, so the difference in

these modes will have a limited impact on overnight nitrate production. We have, however, edited the following lines to 1) acknowledge the two $\gamma(N_2O_5)$ modes in Figure 8, 2) acknowledge the large variability in Figure 9, and 3) state that this variability in Figure 9 (nitrate production rates) is the result of the large variability in the observed nitrate radical production rates ($P_{NO_3,max}$) in Figure 6.

*Page 10, Line 24*
*For the SLV alone (N = 1030), the distribution in **Error! Reference source not found.** shows that $\gamma(N_2O_5)$ values ranged four orders of magnitude from 1 $\times10^{-3}$ to > 1 with two modes centered near 0.01 and 0.08.*

*Page 14, Line 5*
***Error! Reference source not found.** shows the distribution of nightly nitrate production predicted by base case simulations (N = 1033), ranging from ~0 to 31 $\mu g\ m^{-3}$ nitrate night$^{-1}$, with a median of 9.9 $\mu g\ m^{-3}$ nitrate night$^{-1}$.*

*Page 14, Line 17*
*As a result, the large variability in predicted nitrate production rates is reflective of the variability in the observed $NO_3$ radical production rates (Figure 6).*

*Page 14, Line 23*
*Estimating the impact of dilution by including a single first order dilution rate constant ($k_{dilution}$) of 1.3$\times10^{-5}\ s^{-1}$ reduced the median nocturnal nitrate production rate by 42% to 5.7 $\mu g\ m^{-3}$ night$^{-1}$ and resulted in a smaller range (~0 to 16 $\mu g\ m^{-3}$ night$^{-1}$) relative to base case simulations in Figure 9.*

P14: The discussion of dilution/entrainment could be enhanced. The Womak paper was just published on 4/8, making it available. It seems that the authors here are arguing for an entrainment rate that is largely independent of time of day, except for the issue of changing height of the mixed layer. However, one might expect the entrainment rate to differ notably between the daytime and nighttime. It is unclear whether the authors are applying a daytime entrainment rate to the nighttime or really, in general, how entrainment is being accounted for. The origin of the 40% scaling factor is somewhat mysterious as well.

We have attempted to improve the clarity of the manuscript regarding the derivation and application of the $k_{dilution}$ term. To answer the referee's specific questions, Womack et al. 2019 derived a single $k_{dilution}$ value (8x10$^{-6}$ s$^{-1}$) to account for dilution (entrainment) in the boundary layer over the entire multi-day pollution build-up event. A single value was derived due to a lack of observational constraints relating to dilution. Though not directly detailed in the manuscript, Womack et al. increased this dilution (entrainment) rate constant by 40% (1.3x10$^{-5}$ s$^{-1}$) for their simulations of the nocturnal RL to account for the reduced volume of the RL relative to the total volume of the mixed boundary layer. While entrainment rates may vary between day and night conditions, the method of Womack et al. represents the single number that would best represent the average rate during this pollution episode. Therefore, we follow the same procedure and scale the boundary layer dilution rate constant from Womack et al. by 40% (8x10$^{-6}$ /0.6) to estimate the role of dilution/mixing processes on nitrate produced overnight (~14 hours) in the RL during the same pollution event (e.g. Figure 9).

We had originally included a description of $k_{dilution}$ in supplemental section S1.4.1 but have added further details. It now reads:

*Section S1.1.4*
*The dilution rate constant was derived by Womack et al. (2019) as the rate constant that, in combination with the derived surface albedo, allowed an observationally-constrained box model to best reproduce the build-up of total $O_x$ (= $NO_2$ + $O_3$ + 1.5*($HNO_3$ + $pNO_3^-$) + 3*$N_2O_5$ + $ClNO_2$ + PANs + OH + 2*alkyl nitrates) observed between 28 and 31 January, 2017 at the UU ground site. Womack et al. (2019) derived a $k_{dilution}$ value of 8$\times10^{-6}$ s$^{-1}$ for the boundary layer following this approach. Due to the reduced volume of the nocturnal RL relative to the boundary layer, this*

*rate constant was scaled up at night by 40% to maintain constant dilution over the entire pollution build-up period. The same approach was applied to our analysis, which resulted in a $k_{dilution}$ value of $1.3\times10^{-5}$ s$^{-1}$ for the RL. The box model-predicted nocturnal nitrate production rate was most sensitive to this parameter, with a 42.2% reduction in the median predicted rate when including an overnight dilution rate of $1.3\times10^{-5}$ s$^{-1}$. Based on Figure S10 in Womack et al. (2019), the RL dilution rate constant could have reasonably ranged between 1.2 and $2.5\times10^{-5}$ s$^{-1}$ ($0.7 -1.5\times10^{-5}$ s$^{-1}$ / 0.6), depending on the surface albedo. Results incorporating this range of estimated dilution rate constants are discussed further in Section 3.3.3 of the main text and below in Figure S7.*

The main text has been adjusted as follows:

*Page 14, Line 25*
*As described in Womack et al. (2019) (and in Section S1.4.1), a single 1$^{st}$-order dilution rate constant of $8\times10^{-6}$ s$^{-1}$ was derived for pollution event #4 in the SLV by fitting a box model to best reproduce the day-to-day build-up of observed $O_{x,total}$ between 28 January and 1 February at the UU ground site. In the model described by Womack et al. (2019), this rate constant was then scaled up by 40% when simulating the nocturnal RL in order to maintain constant dilution and account for the reduced volume relative to the mixed daytime boundary layer. While dilution / entrainment rates may vary day to night, the method of Womack et al. (2019) represents the single number that would best represent the average rate. The same procedure is followed here with a resulting $k_{dilution}$ value of $1.3\times10^{-5}$ s$^{-1}$, which is ~60% lower than $k_{dilution}$ from the WINTER campaign, derived from observations of $NO_y$ (= $NO + NO_2 + NO_3 + 2*N_2O_5 + ClNO_2 + RONO_2...$) overnight in a single RL air parcel over the eastern U.S. coast (McDuffie et al., 2018b).*

See our response to the Pg 15/Ln 4 comment below for clarification on daytime dilution rate constant.

P15/L3: The difference between the L and M cases seems negligible, as is evident from Fig. 10. Why include both of these when they are so similar?

We have included both estimates because we wanted to test the entire range of possible dilution rate constants, determined by Womack et al., 2019. Even though the L and M dilution rate constant estimates are similar, we have decided to retain both, but have moved them to a new supplemental figure S7 and have updated the main text accordingly.

*Page 15, Line 30*
*When considering the entire possible range of dilution rate constants from Womack et al. (2019), the median values from both cases were between 1.1 and 4.2 $\mu$g m$^{-3}$ day$^{-1}$, as shown in Figure S7.*

*Figure S7.*

[Figure]

Figure S7. (a) For pollution event #4, comparison of model-predicted nocturnal nitrate production ($\mu g$ $m^{-3}$ $day^{-1}$) for base case simulations (gray), simulations with 24-hours of dilution (blue), and the average daily nitrate build-up observed at HW (red). Dilution cases are for simulations that incorporate nocturnal dilution rate constants of $1.2 \times 10^{-5}$ (L), $1.3 \times 10^{-5}$ (M), and $2.5 \times 10^{-5}$ (H) $s^{-1}$, scaled by 60% during the day. Box and whisker plots show the $10^{th} - 90^{th}$ percentile distributions of each set. The red diamond shows the ground-based build-up rate, calculated from 24-hr averaged data at HW in panel b. Upper-limit values assume morning mixing between equivalent nitrate concentrations produced in the RL and NBL. Lower-limit values assume morning mixing with no nitrate production in the NBL (b) Observed concentrations and average daily build-up rate of nitrate aerosol mass (total mass * 0.58) at HW during event #4.

P15/L4: Should the transition from nighttime to daytime also be accounted for? In other words, if most nitrate is formed aloft (as suggested) and the air in the RL is entrained to the surface starting at sunrise, then the nitrate in the RL will be distributed throughout the daytime mixed layer. Dilution from exchange with the FT will occur on top of this. However, that would also require accounting for the nitrate in the surface layer initially.

The referee is correct that morning mixing between the RL and nocturnal boundary layer (NBL) should have been included in addition to daytime dilution from free tropospheric entrainment. Since this box model predicts the amount of nocturnal nitrate produced in the RL only, we have incorporated the effect of morning mixing using two upper and lower limit case estimates. In both cases, the dilution associated with free tropospheric entrainment is treated the same way as originally described: a loss rate constant of $1.3 \times 10^{-5}$ $s^{-1}$ is applied overnight (~14 hours), reduced to $8 \times 10^{-6}$ $s^{-1}$ for the remaining 10 hours in the mixed boundary layer. We also assume instantaneous morning mixing following Womack et al., 2019. The time of mixing will not impact our estimates as we are only simulating nitrate loss processes after sunrise.

First, for the upper limit case, we assume that the amount of nitrate produced overnight in the RL is the same as in the NBL. This will result in the same nitrate concentration in the morning boundary layer after mixing as that produced overnight in the RL. This is an upper limit estimate because nitrate production in the NBL is expected to be smaller than production in the RL due to $O_3$ titration that reduces the $NO_2$ oxidation rate in the NBL. In confirmation, Womack et al., 2019 found that nitrate production in the NBL was lower than in the RL (Womack et al., Figure S6). This upper limit case represents the maximum amount of nocturnally-produced nitrate that would be observed at the ground when considering the effects of 24-hour dilution. This is also the same as the dilution case that we had originally included.

Second, the lower limit case assumes that no nitrate is produced in the NBL overnight (consistent with Jan. 31 and Feb $1^{st}$ in Figure S6 of Womack et al., 2019). In this case, modeled morning nitrate concentrations are diluted by 40% to account for mixing between the NBL (40%) and RL (60%) volumes. Daytime entrainment then follows as described above.

While there are uncertainties, these two cases better capture the possible range of nocturnal nitrate observed at the surface for an improved comparison with surface observations. In addition, we have added

supplemental figure S7 (see response to previous comment) to show the upper and lower limit case results for the entire range of dilution rate constants that was derived by Womack et al., 2019 (1.2-2.5x10$^{-5}$ s$^{-1}$).

To account for these two cases, the following updates have been made to the main text and Figure 10.

[revised manuscript text omitted]

*Figure 10 Updates*
*Original*                                                    *Updated version*

*Figure 1. (a) For pollution event #4, comparison of model-predicted nocturnal nitrate production ($\mu g\ m^{-3}\ day^{-1}$) for base case simulations (gray), simulations with 24-hours of dilution (blue), and the average daily nitrate build-up observed at HW (red). Dilution cases are for simulations that incorporate nocturnal dilution rate constants of $1.2 \times 10^{-5}$ (L), $1.3 \times 10^{-5}$ (M), and $2.5 \times 10^{-5}$ (H) $s^{-1}$, scaled by 60% during the day. Box and whisker plots show the $10^{th} - 90^{th}$ percentile distributions of each set. Upper-limit (UL) values assume morning mixing between equivalent nitrate concentrations produced in the RL and NBL. Lower-limit (LL) values assume morning mixing with no nitrate production in the NBL The red diamond shows the ground-based build-up rate, calculated from 24-hr averaged data at HW in panel b. (b) Observed concentrations and average daily build-up rate of nitrate aerosol mass (total mass \* 0.58) at HW during event #4.*

P15/L9: I have some difficulty with the framing here. The authors start by saying that nocturnal chemistry and largely explain the nitrate accumulation at the surface. But then they go on to say that Womack (who looked at the same events) concluded that photochemical production is quite important too. Notably, while Pusede et al. (2016) implicated nocturnal nitrate production as very important, they did not discount daytime production to the extent suggested here and Prabhakar et al. (2017), building on Pusede et al. (2016), concluded that daytime production plays an important role. I suggest that the authors consider revising the first sentence and how they frame the discussion here. They have only considered nighttime formation and thus their ultimate conclusions regarding the contributions of daytime processes are limited.

We had intended to state that our results suggested that most of the 24-hour nitrate accumulation could be accounted for, not that there was no role of photochemistry. In light of the additional dilution included for morning mixing (discussed above) and these comments, we have made the following changes to this paragraph. We have also changed sentences throughout the text for the same effect.

*Page 15, Line 32*
*Comparison of modeled rates to the observed surface build-up of 4.6 $\mu g\ m^{-3}\ day^{-1}$ suggests that on average, nitrate produced from heterogenous chemistry can account for at least 50% of the nitrate accumulation observed at the surface. This result is qualitatively consistent with an observational analysis by Pusede et al. (2016), who determined that nocturnal heterogeneous chemistry was the main source of regional aerosol nitrate during wintertime pollution events in the San Joaquin Valley. The lower limit estimate, however, is also similar to a box model analysis of this same event by Womack et al. (2019) who found roughly equal contributions between photochemical and nocturnal nitrate production pathways, highlighting that photochemical nitrate production is also occurring during these events. Therefore, while results in Figure 10a (including dilution) predict a median nocturnal fractional contribution of 52 - 86% (ranging between 24 and 91% (Figure S7)), confirmation and further quantification of this result will require additional, vertically resolved measurements of aerosol composition, gas-phase precursors, and physical parameters, as well as more sophisticated modeling of these multi-day pollution accumulation events with 3D-chemical transport models.*

*Page 15, Line 12*
*Direct comparison is difficult, however, as the 24-hour ground-based accumulation rate includes contributions from photochemistry and nocturnal formation in the RL and nocturnal boundary layer (NBL), and is impacted by dilution and mixing processes.*

*Abstract*
*Lastly, additional model simulations suggest nocturnal $N_2O_5$ uptake produces between 2.4 and 3.9 $\mu g$ $m^{-3}$ of nitrate per day when considering the possible effects of dilution. This nocturnal production is sufficient to account for 52 - 86% of the daily observed surface-level build-up of aerosol nitrate, though accurate quantification is dependent on modeled dilution, mixing processes, and photochemistry.*

A minor grammar question: should it be "in the SLV" or "in SLV". I would have thought the former, as the SLV is not a city but a region (like the western US, which the authors refer to with a "the").

We agree and have made this change throughout the main text and supplement.

P13/L5: "empirically-based" should just be "empirical."

Changed

[revised manuscript text omitted]

$^a k_0 = 3.6\times10^{-30}*M*(T/300)^{-4.1}$, $k_\infty = 1.9\times10^{-12}*(T/300)^{0.2}$, KR = $k_0/k_\infty$ NC = 0.75-1.27*$log_{10}$(0.35), F = $10^{(log10(0.35)/(1+log10(KR)/NC)^2))}$, M = Pressure [mbar] $1\times10^{-4}/(k_b*T)$

$^b k_0 = 1.3\times10^{-3}*M*(T/300)^{-3.5}e^{(-11000/T)}$, $k_\infty = 9.7\times10^{14}*(T/300)^{0.1}e^{(-11080/T)}$, KR = $k_0/
[revised manuscript text omitted]

derived a $k_{dilution}$ value of $8\times10^{-6}$ s$^{-1}$ for the boundary layer following this approach. Due to the reduced volume of the nocturnal RL relative to the boundary layer, this rate constant was scaled up at night by 40% to maintain constant dilution over the entire pollution build-up period. The same approach was applied to our analysis, which resulted in a $k_{dilution}$ value of $1.3\times10^{-5}$ s$^{-1}$ for the RL. The box model-predicted nocturnal nitrate production rate was most sensitive to this parameter, with a 42.2% reduction in the median predicted rate when including an overnight dilution rate of $1.3\times10^{-5}$ s$^{-1}$. Based on Figure S10 in Womack et al. (2019), the RL dilution rate constant could have reasonably ranged between 1.2 and $2.5\times10^{-5}$ s$^{-1}$ (0.7 -$1.5\times10^{-5}$ s$^{-1}$ / 0.6), depending on the albedo. Results incorporating this range of estimated dilution rate constants 
[revised manuscript text omitted]

**Formatted Table**

**Section S5 Vertical Profiles**

The vertical profiles of measured N$_2$O$_5$ and ClNO$_2$ and box model-derived γ(N$_2$O$_5$) and φ(ClNO$_2$) values are shown in Figure S6.

[Figure]

**Figure S6. Vertical profiles of N$_2$O$_5$, ClNO$_2$ (1-second measurements), and box-model derived γ(N$_2$O$_5$) and φ(ClNO$_2$) values from all night flights over the SLV. In each panel, light shaded regions show the 10$^{th}$-90$^{th}$ percentile ranges, dark shaded regions are the 25$^{th}$-75$^{th}$ percentile ranges, and the solid lines are the 50$^{th}$ percentile. Dashed black lines show the number of points at each altitude.**

**Section S6 Additional Dilution Results**

As described in Section S1.4.1, Womack et al. (2019) derived a range of rate constants between $0.7\times10^{-5}$ and $1.5\times10^{-5}$ s$^{-1}$ (depending on surface albedo) that could best reproduce the build-up of $O_{x,total}$ observed during pollution event #4. To compare the box model results to the observed ground-based nitrate accumulation rate during the same event, and to assess the role of dilution, Figure 10 from the main text is reproduced here (M), with additional results using nocturnal $k_{dilution}$ rate constants of $1.2\times10^{-5}$ s$^{-1}$ (L) and $2.5\times10^{-5}$ s$^{-1}$ (H) ($0.7\times10^{-5}$ and $1.5\times10^{-5}$ s$^{-1}$ during the day).

[Figure]

**Figure S7.** (a) For pollution event #4, comparison of model-predicted nocturnal nitrate production (µg m$^{-3}$ day$^{-1}$) for base case simulations (gray), simulations with 24-hours of dilution (blue), and the average daily nitrate build-up observed at HW (red). Dilution cases are for simulations that incorporate nocturnal dilution rate constants of $1.2\times10^{-5}$ (L), $1.3\times10^{-5}$ (M), and $2.5\times10^{-5}$ s$^{-1}$ (H), scaled by 60% during the day. Box and whisker plots show the 10$^{th}$ – 90$^{th}$ percentile distributions of each set. The red diamond shows the ground-based build-up rate, calculated from 24-hr averaged data at HW in panel b. Upper-limit values assume morning mixing between equivalent nitrate concentrations produced in the RL and NBL. Lower-limit values assume morning mixing with no nitrate production in the NBL (b) Observed concentrations and average daily build-up rate of nitrate aerosol mass (total mass * 0.58) at HW during event #4.